# High efficiency planar-type perovskite solar cells with negligible hysteresis using EDTA-complexed SnO$_2$

Dong Yang[1,2], Ruixia Yang[1], Kai Wang [2], Congcong Wu[2], Xuejie Zhu[1], Jiangshan Feng[1], Xiaodong Ren[1], Guojia Fang [3], Shashank Priya [2] & Shengzhong (Frank) Liu[1,4]

Even though the mesoporous-type perovskite solar cell (PSC) is known for high efficiency, its planar-type counterpart exhibits lower efficiency and hysteretic response. Herein, we report success in suppressing hysteresis and record efficiency for planar-type devices using EDTA-complexed tin oxide (SnO$_2$) electron-transport layer. The Fermi level of EDTA-complexed SnO$_2$ is better matched with the conduction band of perovskite, leading to high open-circuit voltage. Its electron mobility is about three times larger than that of the SnO$_2$. The record power conversion efficiency of planar-type PSCs with EDTA-complexed SnO$_2$ increases to 21.60% (certified at 21.52% by Newport) with negligible hysteresis. Meanwhile, the low-temperature processed EDTA-complexed SnO$_2$ enables 18.28% efficiency for a flexible device. Moreover, the unsealed PSCs with EDTA-complexed SnO$_2$ degrade only by 8% exposed in an ambient atmosphere after 2880 h, and only by 14% after 120 h under irradiation at 100 mW cm$^{-2}$.

[1] Key Laboratory of Applied Surface and Colloid Chemistry, Ministry of Education; Shaanxi Engineering Lab for Advanced Energy Technology, School of Materials Science and Engineering, Shaanxi Normal University, Xi'an 710119, China. [2] Center for Energy Harvesting Materials and System (CEHMS), Virginia Tech, Blacksburg, VA 24061, USA. [3] Key Laboratory of Artificial Micro- and Nano-structures of Ministry of Education of China, School of Physics and Technology, Wuhan University, Wuhan 430072, China. [4] Dalian National Laboratory for Clean Energy, iChEM, Dalian Institute of Chemical Physics, Chinese Academy of Sciences, 457 Zhongshan Road, Dalian 116023, China. Correspondence and requests for materials should be addressed to D.Y. (email: dongyang@vt.edu) or to S.P. (email: spriya@vt.edu) or to S.L. (email: szliu@dicp.ac.cn)

Owing to the singular properties, including tuned band gap, small exciton energy, excellent bipolar carrier transport, long charge diffusion length, and amazingly high tolerance to defects[1–7], organometal halide perovskites have been projected to be promising candidates for a multitude of optoelectronic applications, including photovoltaics, light emission, photodetectors, X-ray imaging, lasers, gamma-ray detection, subwavelength photonic devices in a long-wavelength region, etc.[8–14]. The rapid increase efficiency in a solar cell based on organometal halide perovskites validates its promise in photovoltaics. In the last few years, the power conversion efficiency (PCE) of mesoporous-type perovskite solar cells (PSCs) has increased to 23.3% by optimizing thin-film growth, interface, and absorber materials[15–17]. As of today, almost all PSCs with high PCE are based on mesoporous-type PSCs that often require high temperature to sinter the mesoporous layer for the best performance, compromising its low-cost advantage and limiting its application in flexible and tandem devices[16,17]. In order to overcome this issue, planar-type PSC comprising of stacked planar thin films has been developed[18,19] using low-temperature and low-cost synthesis processes[20–22] since the long charge diffusion length and bipolar carrier properties of perovskites[23,24]. However, compared to the mesoporous-type PSC, its planar-type counterpart suffers from significant lower certified PCE[18,25].

In a typical planar-type PSC, the perovskite absorber usually inserts between the electron-transport layer (ETL) and the hole-transport layer (HTL) to achieve inverted p–i–n or regular n–i–p configuration[21]. Generally, the inverted device structure utilizing fullerene ETL displays very low hysteresis, however, it usually yields lower PCE, not to mention that fullerene is very expensive[26,27]. Therefore, research has focused on n–i–p architecture to provide low cost and high efficiency[28,29]. Even though ETL-free planar-type PSCs have been reported[30,31], their highest PCE is only 14.14%, significantly lower than that of the cells with ETL, demonstrating the importance of the ETL in this configuration of PSCs. A suitable ETL should meet some basic requirements for high device efficiency[32]. For instance, decent optical transmittance to ensure enough light is transmitted into the perovskite absorber, matched energy level with the perovskite materials to produce the expected open-circuit voltage ($V_{oc}$), and high electron mobility to extract carriers from the active layer effectively in order to avoid charge recombination, etc. Fast carrier extraction is desired to restrict charge accumulation at the interface due to ion migration for reduced hysteresis in the planar-type PSCs. Thus, emphasis has been on developing high-quality ETLs with suitable energy level and high electron mobility for high PCE devices.

Thus far, $TiO_2$ is still the most widely used ETL in high-efficiency n–i–p planar-type PSCs due to its excellent photoelectric properties[33]. However, the electron mobility of $TiO_2$ ETL is too low (ca. $10^{-4}$ cm$^2$ V$^{-1}$ s$^{-1}$) to match with high hole mobility of commonly used HTLs (ca. $10^{-3}$ cm$^2$ V$^{-1}$ s$^{-1}$), leading to charge accumulation at the $TiO_2$/perovskite interface that causes hysteresis and reduced efficiency[34]. There have been extensive efforts in developing low-temperature $TiO_2$ ETL, such as exploring low-temperature synthesis processes through doping and chemical engineering. The results shown by Tan et al. demonstrate that use of chlorine to modify the $TiO_2$ microstructure at low temperatures provides promising PCE of 20.1%[35]. Recently, $SnO_2$ has been demonstrated as an alternative ETL to replace $TiO_2$, owing to its more suitable energy level relative to perovskite and higher electron mobility. Ke et al. first used $SnO_2$ thin film as an ETL in regular planar-type PSCs and demonstrated a PCE of 16.02% with improved hysteresis[36]. Later, the $SnO_2$–$TiO_2$ (planar and mesoporous) composite layers were developed to enhance the performance of the PSCs[37,38]. It is

noteworthy to mention that $Al^{3+}$-doped $SnO_2$ provides even better performance[39]. Subsequently, a variety of methods, such as solution deposition, atomic layer deposition, chemical bath deposition, etc.[40–42] have been developed for synthesizing $SnO_2$ thin film to improve the performance of planar-type PSCs[43]. Recently, Jiang et al. developed the $SnO_2$ nanoparticles as the ETL and demonstrated a certified efficiency as high as 19.9% with very low hysteresis[21]. However, the PCE of the planar-type PSCs is still lower than that of the mesoporous-type devices likely due to charge accumulation at the ETL/perovskite interface caused by relatively low electron mobility of the ETL[44]. It is expected that better PSC performance will be achieved by increasing electron mobility of the ETLs.

Ethylene diamine tetraacetic acid (EDTA) provides excellent modification of ETLs in organic solar cells owing to its strong chelation function. Li et al. have employed EDTA to passivate ZnO-based ETL and demonstrated improved performance of the organic solar cells[45]. However, when the EDTA–ZnO layer is used in the present perovskite cells, the hydroxyl groups or acetate ligands on the ZnO surface react with the perovskite and proton transfer reactions occur at the perovskite/ZnO interface, leading to poor device performance[46].

In the present work, we realize an EDTA-complexed $SnO_2$ (E-$SnO_2$) ETLs by complexing EDTA with $SnO_2$ in planar-type PSCs to attain PCE as high as 21.60%, and certified PCE reaches to 21.52%, the highest reported value to date for the planar-type PSCs. Owing to the low-temperature processing for E-$SnO_2$, we fabricate flexible PSCs, and the PCE reaches to 18.28%. Besides, the PSCs based on E-$SnO_2$ show negligible hysteresis because of the eliminated charge accumulation at the perovskite/ETL interface. We find that the electron mobility of E-$SnO_2$ increases by about three times compared to that of $SnO_2$, leading to negligible current density–voltage (J–V) hysteresis due to improved electron extraction from the perovskite absorber[21]. Furthermore, we find that $SnO_2$ surface becomes more hydrophilic upon EDTA treatment, which decreases the Gibbs free energy for heterogeneous nucleation, resulting in high quality of the perovskite film.

## Results

**Fabrication and characterization of E-$SnO_2$.** It is well known that EDTA can react with transition metal oxide to form a complex, because it can provide its lone-pair electrons to the vacant d-orbital of the transition metal atom[47]. Thus, EDTA was chosen to modify the $SnO_2$ to improve its performance. Supplementary Fig. 1a describes the chemical reaction that occurred when the $SnO_2$ was treated using the EDTA aqueous solution, resulting in the formation of a five-membered ring chelate. The images of EDTA, $SnO_2$, and E-$SnO_2$ samples are shown in Supplementary Fig. 1b. It is apparent that the unmodified EDTA and $SnO_2$ samples are transparent, while EDTA-treated $SnO_2$ turned into milky white. Supplementary Fig. 2 compares the Fourier-transform infrared spectroscopy (FTIR) spectra of the E-$SnO_2$ solution measured in the freshly prepared condition and again after it was stored in an ambient atmosphere for 2 months. It is clear that there is no obvious difference between the two solutions indicating the high stability.

Figure 1a shows the X-ray photoelectron spectra (XPS) for EDTA, $SnO_2$, and E-$SnO_2$ films deposited on quartz substrates. In order to reduce the charging effect, the exposed surface of the quartz substrate was coated with a conductive silver paint and connected to the ground. We calibrated the binding energy scale for all XPS measurements to the carbon 1s line at 284.8 eV. It is clear from these measurements that $SnO_2$ shows only peaks attributed to Sn and O. After the EDTA treatment, the E-$SnO_2$ film shows an additional peak located at ca. 400 eV, ascribed to N.

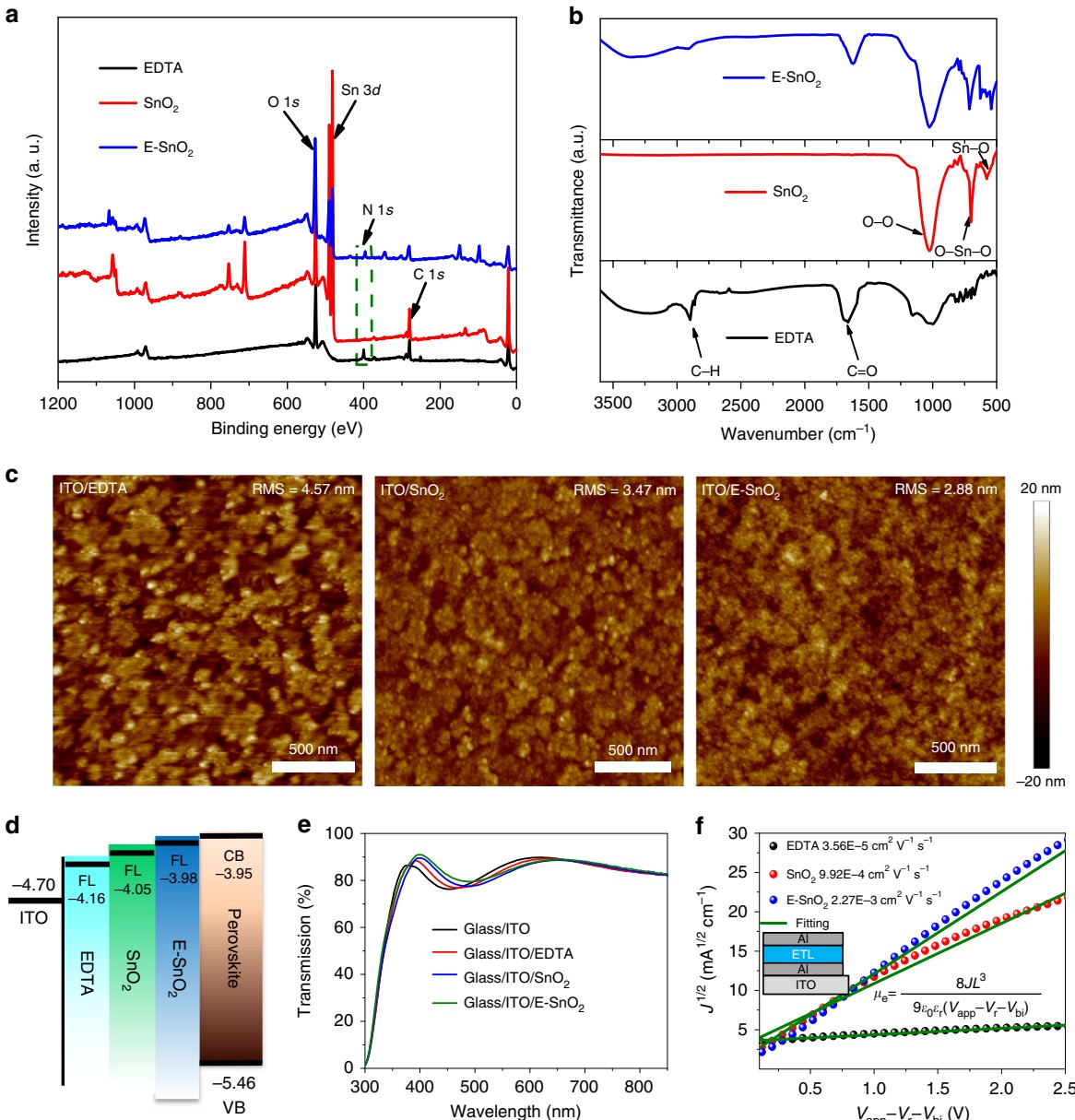

**Fig. 1** Characterization of the ETLs. **a** XPS and **b** FTIR spectra of EDTA, SnO$_2$, and E-SnO$_2$ films deposited on quartz substrates. **c** AFM topographical images of EDTA, SnO$_2$, and E-SnO$_2$ films. **d** Schematic illustration of Fermi level of EDTA, SnO$_2$, and E-SnO$_2$ relative to the conduction band of the perovskite layer. The Fermi level of EDTA, SnO$_2$, and E-SnO$_2$ is measured by KPFM, and the conduction and valence band of the perovskite materials are obtained from the previous report[74]. **e** Optical transmission spectra of EDTA, SnO$_2$, and E-SnO$_2$ films on ITO substrates. **f** Electron mobility for EDTA, SnO$_2$, and E-SnO$_2$ using the SCLC model, and the inset shows the device structure of ITO/Al/ETL/Al

Meanwhile, the Sn 3$d$ peaks from E-SnO$_2$ are shifted by ca. 0.16 eV in contrast to the pristine SnO$_2$ (Supplementary Fig. 3), indicating that EDTA is bound to the SnO$_2$.

FTIR was used to study the interaction between SnO$_2$ and EDTA. As shown in Fig. 1b, the peaks around 2895 cm$^{-1}$ and 1673 cm$^{-1}$ belong to C–H and C=O stretching vibration in the EDTA, respectively. The characteristic peaks of SnO$_2$ observed at ca. 701 cm$^{-1}$ and 549 cm$^{-1}$ are due to O–Sn–O stretch and the Sn–O vibration, respectively[48]. In addition, the peak at 1040 cm$^{-1}$ in the SnO$_2$ film is attributed to O–O stretching vibration due to oxygen adsorption on the SnO$_2$ surface[49]. For the E-SnO$_2$ sample, the characteristic peaks of SnO$_2$ shift to 713 cm$^{-1}$ and 563 cm$^{-1}$, and the C–H and C=O stretching vibration peaks shift to 2913 cm$^{-1}$ and 1624 cm$^{-1}$, further demonstrating that the EDTA is indeed complexed with SnO$_2$.

Atomic force microscopy (AFM) images of EDTA, SnO$_2$, and E-SnO$_2$ films deposited on the ITO substrates are shown in Fig. 1c. The data reveal that the E-SnO$_2$ film shows the smallest root-mean-square roughness of 2.88 nm, a key figure-of-merit for the PSCs[50]. We also measured their Fermi level by Kelvin probe force microscopy (KPFM), with the surface potential images shown in Supplementary Fig. 4, and the calculated details are described in Supplementary Note 1. Figure 1d provides energy band alignment between perovskites and different ETLs. The Fermi level of E-SnO$_2$ is very close to the conduction band of perovskite, which is beneficial for enhancing $V_{oc}$[51].

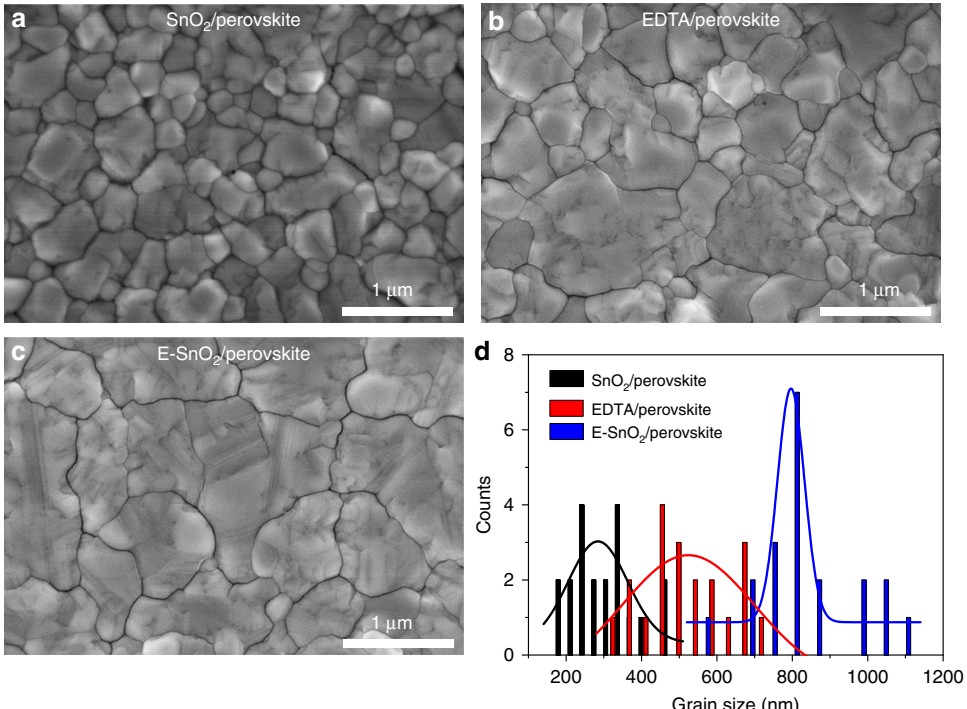

**Fig. 2** The morphology of perovskite films deposited on different substrates. Top-view scanning electron microscope (SEM) images of perovskite films coated on **a** EDTA, **b** SnO$_2$, and **c** E-SnO$_2$ substrates. **d** The grain size distribution of perovskite deposited on various substrates

Figure 1e shows the optical transmission spectra of EDTA, SnO$_2$, and E-SnO$_2$ films coated on ITO. All these samples display high average transmittance in the visible region, demonstrating good optical quality. In addition, the electron mobility of various ETLs was measured using the space charge-limited current (SCLC) method[20], as shown in Fig. 1f. It is found that electron mobility of E-SnO$_2$ is $2.27 \times 10^{-3}$ cm$^2$ V$^{-1}$ s$^{-1}$, significantly larger than those of the EDTA ($3.56 \times 10^{-5}$ cm$^2$ V$^{-1}$ s$^{-1}$) and the SnO$_2$ ($9.92 \times 10^{-4}$ cm$^2$ V$^{-1}$ s$^{-1}$). It is known that the electron mobility is a key figure-of-merit for ETLs in PSCs. Supplementary Fig. 5 shows the electron injection models for ITO/SnO$_2$ or E-SnO$_2$/perovskite/PCBM/Al structures, with their corresponding J–V curves, and the details are described in Supplementary Note 2. It is apparent that the high electron mobility effectively promotes electron transfer in the PSCs, reduces charge accumulation at the ETL/perovskite interface, improves efficiency, and suppresses hysteresis for the PSCs[21].

**Perovskite growth mechanism.** The quality of the perovskite films, including grain size, crystallinity, surface coverage, etc., is very important for high-performance PSCs. For a consistent microstructure, a solution deposition technique was used to fabricate perovskite films on EDTA, SnO$_2$, and E-SnO$_2$ substrates. Figure 2a–c shows the morphology of the perovskite films deposited on different ETLs. It is clear from these images that continuous pinhole-free films with full surface coverage were obtained. Figure 2d shows the distribution diagram with an average grain size of about 309 nm for the perovskite coated on SnO$_2$. The grain size increased to about 518 nm for the EDTA sample. Surprisingly, the average perovskite grain size is further enhanced to as much as about 828 nm (Fig. 2c, d) for the E-SnO$_2$ substrates.

According to the established model for nucleation and growth of thin films[52,53], the perovskite formation process can be divided into four steps: (i) formation of a crystal nucleus, (ii) evolution of nuclei into an island structure, (iii) formation of a networked

microstructure, and (iv) growth of networks into a continuous film. The Gibbs free energy for heterogeneous nucleation in the first step can be expressed as Eq. (1)

$$\triangle G_{\text{heterogeneous}} = \triangle G_{\text{homogeneous}} \times f(\theta) \qquad (1)$$

wherein $f(\theta) = (2-3\cos\theta + \cos^3\theta)/4$[54], and $\theta$ is the contact angle of the precursor solution. Since the magnitude of $\theta$ varies in the range of [0, $\pi$/2], the larger the $\theta$ is, the smaller is the magnitude of $\cos\theta$, and therefore larger is the parameter $f(\theta) \in$ [0, 1]. In other words, a smaller contact angle results in reduced Gibbs free energy for heterogeneous nucleation, thereby assisting the nucleation process. Higher nucleation density will promote the film densification process[53]. Compared to EDTA and SnO$_2$, E-SnO$_2$ shows the smallest contact angle (20.67°, Supplementary Fig. 6), resulting in the wettability interface for the perovskite[55–57]. Thus, the perovskite coated on the E-SnO$_2$ exhibits better crystallinity (Supplementary Fig. 7) and full surface coverage (Fig. 2c). In addition, the small contact angle of the substrate provides the low surface energy[58], leading to increased grain size during the growth of the networked structure[53], as observed in the SEM measurements.

**Charge transfer dynamics.** The electron-only devices with the structure of ITO/ETL/perovskite/PCBM/Ag were fabricated to evaluate the trap density of perovskite deposited on different substrates. Figure 3a shows the dark current–voltage (I–V) curves of the electron-only devices. The linear correlation (dark yellow line) reveals an ohmic-type response at low bias voltage, when the bias voltage is above the kink point, which defines as the trap-filled limit voltage ($V_{\text{TFL}}$), the current nonlinearly increases (cyan line), indicating that the traps are completely filled. The trap density ($N_t$) can be obtained using Eq. (2)

$$N_t = \frac{2\varepsilon_0 \varepsilon V_{\text{TFT}}}{eL^2} \qquad (2)$$

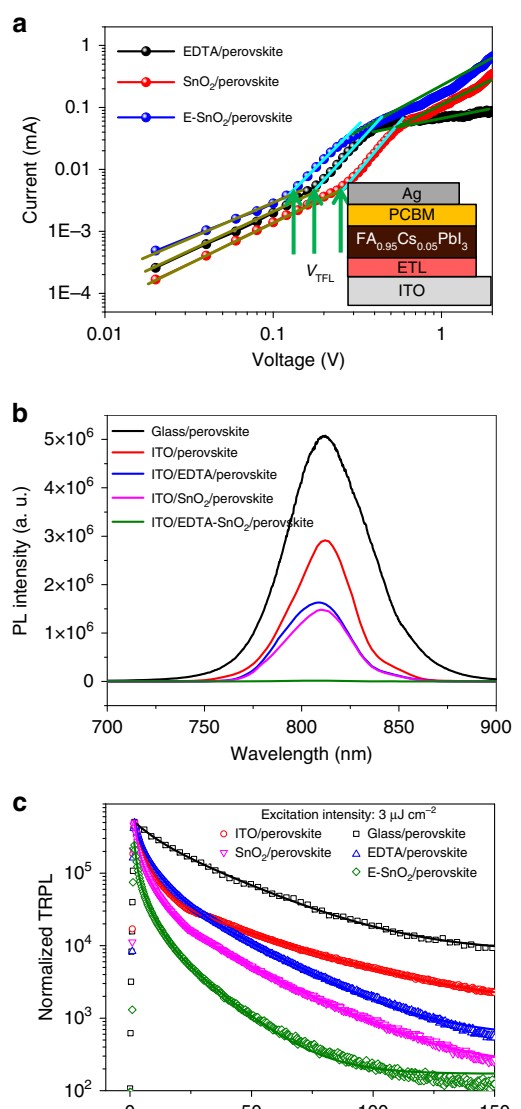

**Fig. 3** The charge transfer between perovskite and different ETLs. **a** Dark *I–V* curves of the electron-only devices with the $V_{TFL}$ kink points. The inset shows the structure of the electron-only device. **b** Steady-state PL and **c** TRPL spectra with an excitation intensity of $3\,\mu J\,cm^{-2}$ of perovskite films deposited on different substrates

where $\varepsilon_0$ is the vacuum permittivity, $\varepsilon$ is the relative dielectric constant of $FA_{0.95}Cs_{0.05}PbI_3$ ($\varepsilon = 62.23$)[59], $e$ is the electron charge, and $L$ is the thickness of the film. The trap densities of the perovskite film coated on $SnO_2$ and EDTA substrates are $1.93 \times 10^{16}$ and $1.27 \times 10^{16}\,cm^{-3}$, respectively. Interestingly, the trap density is reduced to as low as $8.97 \times 10^{15}\,cm^{-3}$ for the film deposited on E-$SnO_2$. The significantly lower trap density is related to low grain boundary density in the perovskite film (Fig. 2).

Figure 3b shows the steady-state photoluminescence (PL) spectra of the perovskite deposited on different substrates. Compared with other samples, significant PL quench is observed in the ITO/E-$SnO_2$/perovskite, demonstrating that the E-$SnO_2$ has the most appealing merits as the highest electron mobility (Fig. 1f). Figure 3c shows the normalized time-resolved PL (TRPL) for perovskite coated on various ETLs. The lifetime and the corresponding amplitudes are listed in Supplementary Table 1. Generally, the slow decay component ($\tau_1$) is attributed to the radiative recombination of free charge carriers due to traps in the

bulk, and the fast decay component ($\tau_2$) is originated from the quenching of charge carriers at the interface[60]. The glass/perovskite sample shows the longest lifetime under excitation intensity of $3\,\mu J\,cm^{-2}$. For perovskite coated on the ITO substrate, the lifetime is decreased to more than half due to the charge transfer from perovskite into ITO. For EDTA/perovskite and $SnO_2$/perovskite samples, both the fast and slow decay lifetimes are very similar, and $\tau_1$ dominates the PL decay for both samples, indicating severe recombination before they were extracted. When the perovskite is deposited on E-$SnO_2$, both $\tau_1$ and $\tau_2$ were shortened to 14.16 ns and 0.97 ns, with a proportion of 45.32% and 54.68%, respectively. Meanwhile, $\tau_2$ appears to dominate the PL decay, indicating that electrons are effectively extracted from the perovskite layer to the E-$SnO_2$ with minimal recombination loss. Even under smaller excitation intensity (0.5 $\mu J\,cm^{-2}$), the acceleration of the lifetime for E-$SnO_2$/perovskite is observed. The lifetime increases with reduced excitation intensity (Supplementary Fig. 8 and Supplementary Table 1), in agreement with a previous report[61]. The electron-transport yield ($\Phi_{tr}$) of different ETLs with different excitation intensities can be estimated using equation, $\Phi_{tr} = 1 - \tau_p/\tau_{glass}$, where $\tau_p$ is the average lifetime for perovskite deposited on different substrates, and $\tau_{glass}$ is the average lifetime for glass/perovskite. With the excitation intensity of $3\,\mu J\,cm^{-2}$, the electron-transport yields of ITO, EDTA, $SnO_2$, and E-$SnO_2$ are 49.72%, 67.58%, 68.31%, and 81.50%, respectively. When the excitation intensity reduces to 0.5 $\mu J\,cm^{-2}$, the electron-transport yields of ITO, EDTA, $SnO_2$, and E-$SnO_2$ are increased to 60.37%, 74.46%, 80.65%, and 90.82%, respectively. It is clear that the excitation intensity can significantly increase the electron-transport yield. These results further indicate that the E-$SnO_2$ is a good electron extraction layer for planar-type PSCs.

**The performance of PSCs**. With the superior optoelectronic properties discussed above, it is expected that the E-$SnO_2$ would make a better ETL in the PSCs than the $SnO_2$. Planar-type PSCs are therefore designed and fabricated based on different ETLs with the device structure shown in Fig. 4a inset. $FAPbI_3$ was used as the active absorber for its proper band gap, with a small amount of Cs doping to improve its phase stability[62,63]. Supplementary Fig. 9 presents the cross-sectional SEM images for the complete device structure. The thickness of the perovskite film is controlled at ca. 420 nm for all devices. While the perovskite grains are not large enough to penetrate through the film thickness when the $SnO_2$ is used as the substrate, the grains are significantly larger when deposited on EDTA and E-$SnO_2$ with the grains grown across the film thickness, which is consistent with top-view SEM results (Fig. 2).

Figure 4a shows the *J–V* curves of planar-type PSCs using different ETLs, with the key parameters, including short-circuit current density ($J_{sc}$), $V_{oc}$, fill factor (FF), and PCE summarized in Table 1. The device based on EDTA gives a PCE of 16.42% with $J_{sc}$ = 22.10 mA cm$^{-2}$, $V_{oc}$ = 1.08 V, and FF = 0.687. The device based on $SnO_2$ substrate shows a PCE of 18.93% with $J_{sc}$ = 22.79 mA cm$^{-2}$, $V_{oc}$ = 1.10 V, and FF = 0.755. Interestingly, when the E-$SnO_2$ is employed as ETL, the $J_{sc}$, FF, and $V_{oc}$ are increased to 24.55 mA cm$^{-2}$, 0.792, and 1.11 V, yielding a PCE up to 21.60%, (the certified efficiency is 21.52%, and the certificated document is shown in Supplementary Fig. 10), the highest efficiency reported to date for the planar-type PSCs. The low device performance for the EDTA is caused by small $J_{sc}$ and FF, which is related to low electron mobility and high resistance[47], and the low $V_{oc}$ results from the small offset of Fermi energy between the EDTA and HTL (Fig. 1d)[64]. In comparison, the planar-type PSCs with E-$SnO_2$ ETLs exhibit the best performance. The higher $J_{sc}$ and FF are

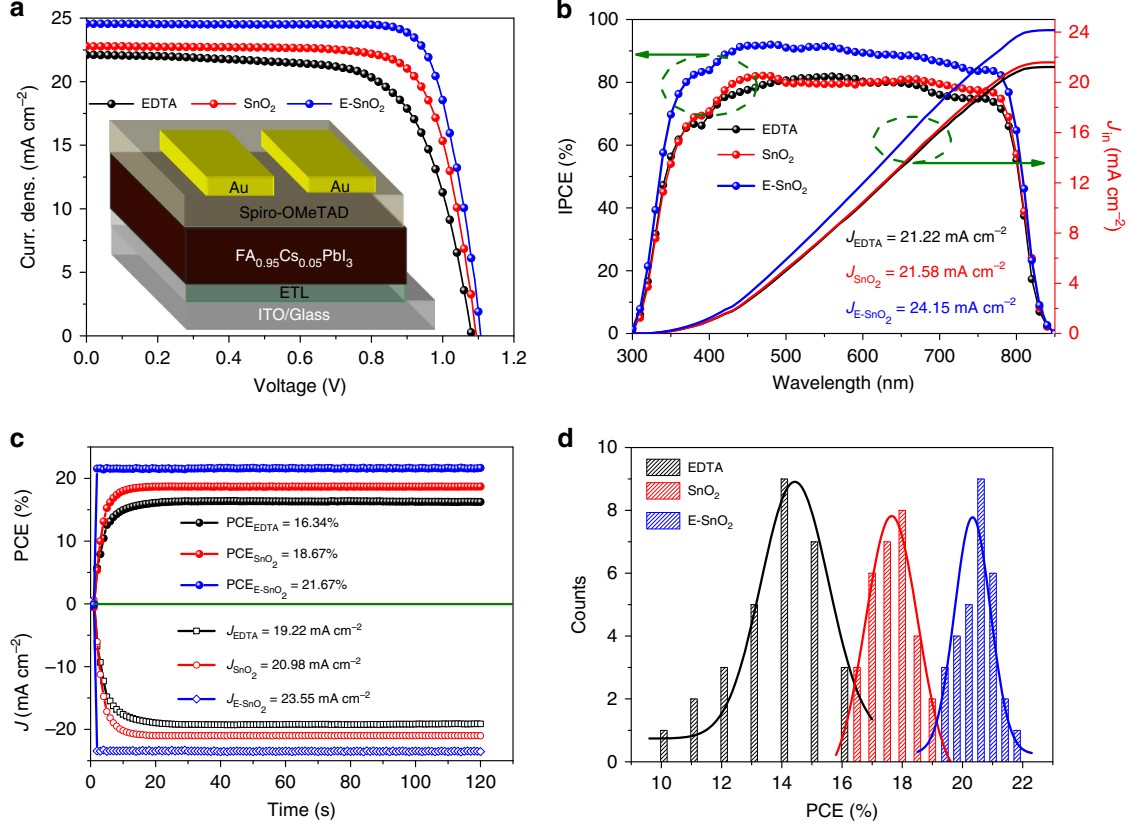

**Fig. 4** PSC performance using ETLs. **a** J–V curves with the inset showing device configuration, and **b** the corresponding IPCE of the planar-type PSCs with various ETLs. The integrated current density from the IPCE curves with the AM 1.5 G photon flux spectrum. **c** Static current density and PCE measured as a function of time for the EDTA, SnO₂, and E-SnO₂ devices biased at 0.85 V, 0.89 V, and 0.92 V, respectively. **d** The PCE distribution histogram of the planar-type PSCs based on different ETLs

| Table 1 The parameters of the rigid and flexible devices | | | | | |
|---|---|---|---|---|---|
| **Style** | **ETL** | $J_{sc}$ (mA cm$^{-2}$) | $V_{oc}$ (V) | **FF** | **PCE (%)** |
| Rigid | EDTA | 22.10 | 1.08 | 0.687 | 16.42 |
| | | 21.43 ± 1.19 | 1.05 ± 0.02 | 0.649 ± 0.074 | 14.60 ± 1.60 |
| | SnO₂ | 22.79 | 1.10 | 0.755 | 18.93 |
| | | 22.70 ± 0.32 | 1.08 ± 0.03 | 0.735 ± 0.022 | 18.04 ± 0.63 |
| | E-SnO₂ | 24.57 | 1.11 | 0.792 | 21.60 |
| | | 24.55 ± 0.76 | 1.11 ± 0.01 | 0.750 ± 0.011 | 20.41 ± 0.55 |
| Flexible | E-SnO₂ R₀ | 23.42 | 1.09 | 0.716 | 18.28 |
| | | 22.64 ± 0.46 | 1.09 ± 0.03 | 0.699 ± 0.028 | 17.26 ± 0.75 |
| | E-SnO₂ R₁₄-500 | 23.42 | 1.09 | 0.715 | 18.25 |
| | E-SnO₂ R₁₂-500 | 23.11 | 1.08 | 0.714 | 17.82 |
| | E-SnO₂ R₇-500 | 22.66 | 1.08 | 0.688 | 16.84 |

attributed to the high electron mobility that promotes effective electron extraction, and the larger $V_{oc}$ due to the closer energy level between E-SnO₂ and perovskite[65]. Figure 4b shows the incident-photon-to-charge conversion efficiency (IPCE) and the integrated current of the PSCs based on different ETLs. The integrated current values calculated by the IPCE spectra for the devices using EDTA, SnO₂, and E-SnO₂ are 21.22, 21.58, and 24.15 mA cm$^{-2}$, respectively, very close to the J–V results. It is apparent that the device based on the E-SnO₂ shows significantly higher IPCE due to less optical loss when perovskite is deposited on E-SnO₂ ETL (Supplementary Fig. 11), consistent with the J–V measurement.

To further demonstrate the device characteristics, photocurrent density of the champion devices from each group based on EDTA, SnO₂, and E-SnO₂ was measured when the devices were biased at 0.85, 0.89, and 0.92 V, respectively. Figure 4c shows the corresponding curves at the maximum power point ($V_{mp}$) in the J–V plots. The PCEs of the champion devices using the EDTA, SnO₂, and E-SnO₂ stabilize at 16.34%, 18.67%, and 21.67% with photocurrent densities of 19.22, 20.98, and 23.55 mA cm$^{-2}$, respectively, very close to the values measured from the J–V curves. Next, we fabricated and measured 30 individual devices for each ETL to study repeatability. Figure 4d shows the PCE distribution histogram for devices with different ETLs, with the statistics listed in Supplementary Tables 2–4. Amazingly, the devices based on E-SnO₂ exhibit excellent repeatability with a very small standard deviation in contrast to the devices based on EDTA and SnO₂, indicating that the E-SnO₂ is an excellent ETL in the planar-type PSC.

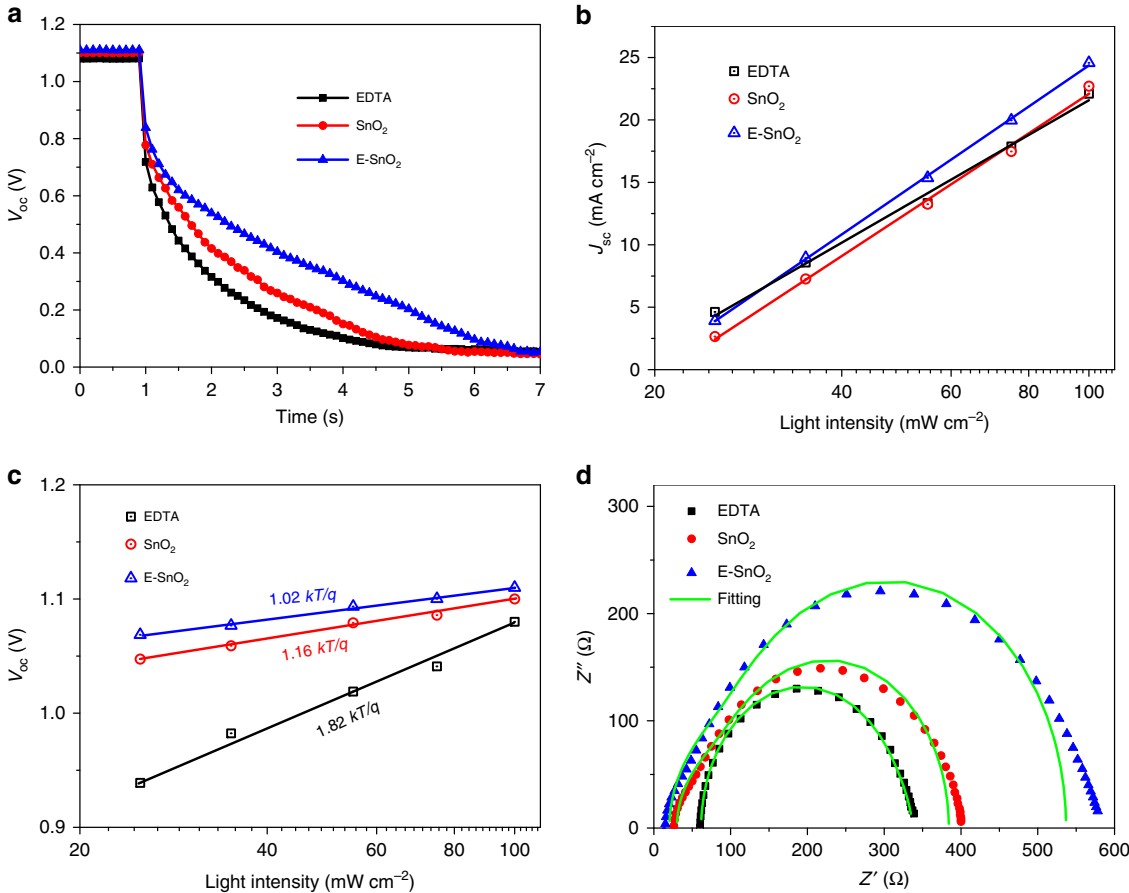

**Fig. 5** Charge transfer properties of the planar-type PSCs using different ETLs. **a** $V_{oc}$ decay curves, **b** $J_{sc}$ vs. light intensity, **c** $V_{oc}$ vs. light intensity, and **d** EIS of planar-type PSCs with various ETLs

In order to gain further insight into the charge transport mechanism, the charge transfer processes in the perovskite devices were studied in detail. The carrier recombination rate in the PSCs was evaluated by the $V_{oc}$ decay measurements. Figure 5a shows the $V_{oc}$ decay curves of the PSCs based on different ETLs. It is apparent that the planar-type PSC based on E-SnO$_2$ exhibits the slowest $V_{oc}$ decay time compared to the devices based on EDTA and SnO$_2$, indicating that the devices with E-SnO$_2$ have the lowest charge recombination rate and the longest carrier lifetime, consistent with the highest $V_{oc}$ for the device based on E-SnO$_2$ by $J$–$V$ measurements. Figure 5b shows $J_{sc}$ versus light intensity of the PSCs using various ETLs. It appears that all devices show a linear correlation with the slopes very close to 1, indicating that the bimolecular recombination in the devices is negligible[66]. Figure 5c shows that $V_{oc}$ changes linearly with the light intensity. Prior studies have indicated that the deviation between the slope and the value of $(kT/q)$ reflects the trap-assisted recombination[20]. In the present case, the device using the E-SnO$_2$ shows the smallest slope, indicating the least trap-assisted recombination, which is in excellent agreement with the result showing the lowest trap density when the perovskite is deposited on E-SnO$_2$ (Fig. 3a). In fact, the slope is as small as 1.02 $kT/q$, implying that the trap-assisted recombination is almost negligible.

The electrical impedance spectroscopy (EIS) was employed to extract transfer resistance in the solar cells. Figure 5d shows the Nyquist plots of the devices using different ETLs measured at $V_{oc}$ under dark conditions, with the equivalent circuit shown in Supplementary Fig. 12. It is known that in the EIS analysis, the high-frequency component is the signature of the transfer resistance ($R_{tr}$) and the low-frequency one for the recombination

resistance ($R_{rec}$)[67]. In the present study, because the perovskite/HTL interface is identical for all devices, the only variable affecting $R_{tr}$ is the perovskite/ETL interface. The numerical fitting gives the device parameters, as listed in Supplementary Table 5. Apparently, compared to PSCs based on EDTA and SnO$_2$, the device with E-SnO$_2$ shows the smallest $R_{tr}$ of 14.8 Ω and the largest $R_{rec}$ of 443.3 Ω. The small $R_{tr}$ is beneficial for electron extraction, and the large $R_{rec}$ effectively resists charge recombination, which is in agreement with the observations discussed above. Combined, all the results confirm that E-SnO$_2$ is the most effective ETL for the planar-type PSC.

**Stability and hysteresis**. Stability and hysteresis are two key characteristics for the PSCs. Figure 6a shows normalized PCE measured as a function of storage time, with more detailed $J$–$V$ parameters summarized in Supplementary Table 6. It is clear that while the device based on E-SnO$_2$ maintains 92% of its initial efficiency exposed to an ambient atmosphere after 2880 h in the dark, the device using SnO$_2$ only provides 74% of its initial efficiency under the same storage condition. The PSCs were also tested under continuous irradiation at 100 mW cm$^{-2}$. Figure 6b shows the normalized PCE changes as a function of test time, with more detailed $J$–$V$ parameters provided in Supplementary Table 7. It is clear that after 120 h of illumination, the device using the E-SnO$_2$ maintains 86% of its initial efficiency, while for the same test duration, the device using SnO$_2$ remains only 38% relative to its initial efficiency. It is apparent that the device fabricated on E-SnO$_2$ shows excellent stability under both the dark and continuous irradiation. The instability of PSC is mainly

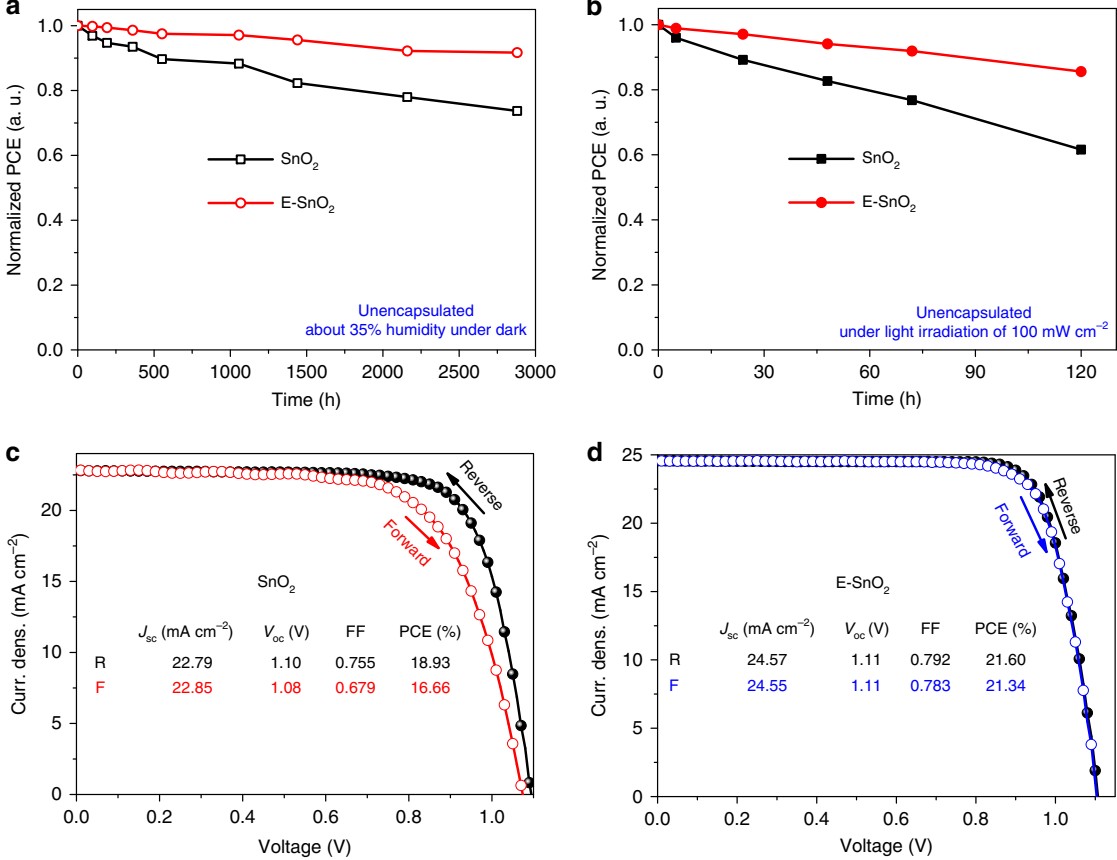

**Fig. 6** Stability and hysteresis test for planar-type PSCs. Long-term stability measurements of devices without any encapsulation under **a** ambient condition and **b** illumination of 100 mW cm$^{-2}$. The J–V curves of the device with **c** SnO$_2$ and **d** E-SnO$_2$ measured under both reverse- and forward-scan directions

caused by degradation of the perovskite film and spiro-OMeTAD HTL. In the present work, all devices used the same spiro-OMeTAD HTL, therefore, the degradation from the HTL should be the same for all the devices. It is found that the grain size of the perovskite film is increased by three times when it is deposited on E-SnO$_2$ in comparison to that on the pristine SnO$_2$ (Fig. 2). The larger grain size can effectively suppress the moisture permeation at grain boundaries[68], resulting in improved environmental stability for the PSCs based on the E-SnO$_2$ ETLs.

For the hysteresis test, Fig. 6c and d show the J–V curves measured under both reverse- and forward- scan directions. It is found that the device with E-SnO$_2$ has almost identical J–V curves with negligible hysteresis, even when it is measured using different scan rates from 0.01 to 0.5 V s$^{-1}$. Supplementary Fig. 13 presents J–V curves measured for the device based on E-SnO$_2$ at different scan rates. It is apparent that the J–V curves almost remain the same, regardless of scan rate and direction, demonstrating that the hysteresis is negligible. Generally, the hysteresis of PSCs is ascribed to interfacial capacitance caused by charge accumulation at the interface, which originates from ion migration, high trap density, and unbalanced charge transport within the perovskite device[69–71]. It is found that the trap density of the perovskite film is significantly reduced when it is deposited on the E-SnO$_2$, one of the primary reasons for reduced hysteresis. In addition, the electron mobility of the SnO$_2$ ETL is only 9.92 × 10$^{-4}$ cm$^2$ V$^{-1}$ s$^{-1}$ (Fig. 1f), about an order of magnitude slower than the hole mobility of the doped spiro-OMeTAD (ca. 10$^{-3}$ cm$^2$ V$^{-1}$ s$^{-1}$) HTL. Thus, the electron flux ($F_e$) is ca. 10 times smaller than the hole flux ($F_h$) due to the same interface area of the ETL/perovskite and perovskite/HTL, that leads to charge accumulation at the SnO$_2$/perovskite interface, as shown in

Supplementary Fig. 14a. The accumulated charge would cause hysteresis in the solar cells (Fig. 6c). When the high electron mobility E-SnO$_2$ (2.27 × 10$^{-3}$ cm$^2$ V$^{-1}$ s$^{-1}$) is employed as the ETL, the $F_e$ is comparable to the $F_h$ of the spiro-OMeTAD HTL (Supplementary Fig. 14b), resulting in equivalent charge transport at both electrodes. Therefore, the high electron mobility of E-SnO$_2$ would enhance electron transport from perovskite to E-SnO$_2$ ETL, leading to no significant charge accumulation, and consequently, the devices based on the E-SnO$_2$ exhibit negligible hysteresis.

**High-efficiency flexible PSCs**. Given the advantage of low-temperature preparation, we applied the E-SnO$_2$ ETL in flexible PSCs. Figure 7a shows J–V curves of flexible PSCs using the poly (ethylene terephthalate) (PET)/ITO substrates, with key J–V parameters summarized in Table 1. The champion flexible device exhibits PCE of 18.28% ($J_{sc}$ = 23.42 mA cm$^{-2}$, $V_{oc}$ = 1.09 V, and FF = 0.716). The lower $J_{sc}$ of the flexible device is caused by the lower transparency of the PET/ITO substrate compared to the glass/ITO used for the rigid device (Supplementary Fig. 15). The lower $V_{oc}$ and FF are likely due to higher sheet resistance of the PET/ITO substrate[67]. Figure 7c shows the IPCE and integral current density of the flexible device. It is clear that the integral current is 23.12 mA cm$^{-2}$, in perfect agreement with the J–V results. For the reproducibility test, 30 individual cells were fabricated with the PCE distribution histogram shown in Fig. 7d and detailed parameters are summarized in Supplementary Table 8, both confirming very good reproducibility.

The mechanical stability is an important quality indicator for the flexible solar cells. According to a previous report[72], it is safe

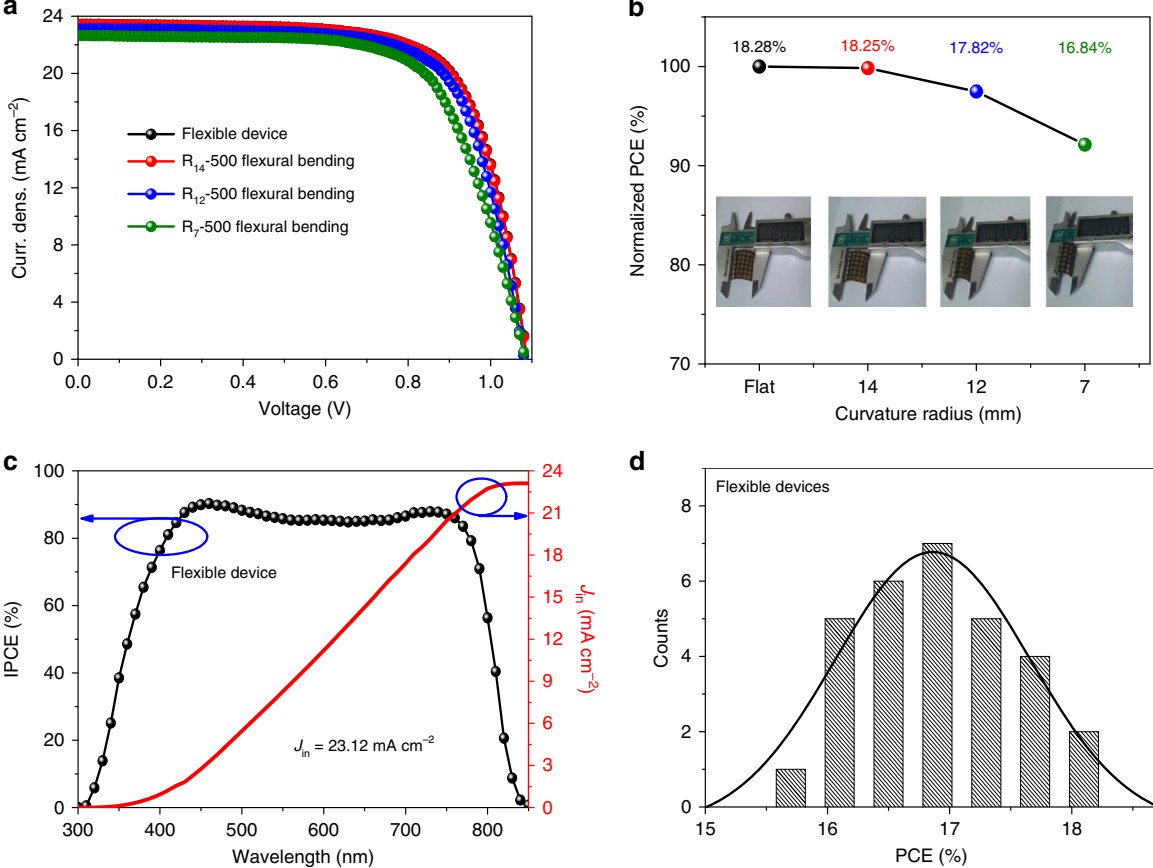

**Fig. 7** The performance of flexible PSCs based on E-SnO$_2$ ETLs. **a** $J$–$V$ curves of the flexible devices and after flexing at curvature radii of 14 mm, 12 mm, and 7 mm for 500 cycles, respectively. **b** The normalized PCE measured after flexing at different curvature radii. **c** IPCE curves of the flexible device. **d** The PCE distribution histogram of the flexible PSCs

for ITO to be bended to a radius of 14 mm, and when the bending radius is smaller than 14 mm, the ITO layer starts to crack, leading to significant degradation in conductivity. In order to examine the mechanical stability of the flexible PSCs, we therefore adopted the bending radii of 14 mm, 12 mm, and 7 mm to test the flexible device. Figure 7a shows device performance of the flexible solar cells measured after flexing for 500 times with different curvature radii, and the test procedure is shown in Fig. 7b. It shows that after flexing for 500 times at a bending radius of 14 mm, the $J$–$V$ curve and the associated parameters remain the same without obvious degradation. However, when the bending radius is decreased to 12 mm and 7 mm, the PCE degraded to 17.82% and 16.84%, respectively, attributing to the conductivity degradation of ITO[72].

## Discussion

An effective E-SnO$_2$ ETL has been developed, and the PCE of planar-type PSCs is increased to 21.60% with negligible hysteresis, and the certified efficiency is 21.52%, this is the highest reported value for planar-type PSCs so far. By taking advantage of low-temperature processing for E-SnO$_2$ ETLs, flexible devices with high PCE of 18.28% are also fabricated. The significant performance of the planar-type PSCs is attributed to the superior advantages when perovskite is deposited on E-SnO$_2$ ETLs, including larger grain size, lower trap density, and good crystallinity. The higher electron mobility facilitates electron transfer for suppressed charge accumulation at the interface, leading to high efficiency with negligible $J$–$V$ hysteresis. Furthermore, the long-term stability is significantly enhanced since the large grain size

that suppressed perovskite degradation at grain boundaries. This work provides a promising direction toward developing high-quality ETLs, and we believe that the present work will facilitate transition of perovskite photovoltaics.

## Methods

**Materials**. NH$_2$CHNH$_2$I (FAI) was synthesized and purified according to a reported procedure[45]. The SnO$_2$ solution was purchased from Alfa Aesar (tin (IV) oxide, 15 wt% in H$_2$O colloidal dispersion). PbI$_2$ (purity > 99.9985%) was purchased from Alfa Aesar. EDTA (purity > 99.995%), CsI (purity > 99.999%), dimethylformamide (DMF, purity > 99%), and dimethyl sulfoxide (DMSO, purity > 99%) were obtained from Sigma Aldrich. In total, 2,2′,7,7′-tetrakis(N,N-di-p-methoxyphenylamine)-9,9′-spirobifluorene (spiro-OMeTAD) was bought from Yingkou OPV Tech Co., Ltd. All of the other solvents were purchased from Sigma Aldrich without any purification.

**Fabrication of EDTA, SnO$_2$, and E-SnO$_2$ films**. The 0.2-mg EDTA was dissolved in 1 mL of deionized water, and the SnO$_2$ aqueous colloidal dispersion (15 wt%) was diluted using deionized water to the concentration of 2.5 wt%. These solutions were stirred at room temperature for 2 h. The SnO$_2$ and EDTA layers were fabricated by spin-coating at 5000 rpm for 60 s using the corresponding solution, and then dried in a vacuum oven at 60 °C under ca. 5 Pa for 30 min to remove residual solvent. The EDTA and SnO$_2$ solution were mixed with a volume ratio of 1:1, then put on a hot plate at 80 °C for 5 h under stirring conditions, and the milky-white E-SnO$_2$ colloidal solution (Supplementary Fig. 1b) was obtained. The E-SnO$_2$ colloidal solution was spin-coated at 5000 rpm for 60 s, and then transferred the samples into a vacuum oven at 60 °C under ca. 5 Pa for 30 min to remove the residual solvent. Finally, the E-SnO$_2$ films were obtained.

**Electron mobility of EDTA, SnO$_2$, and E-SnO$_2$ films**. To gain insights into the charge transport, we have measured electron mobility using different ETLs in the same device structure. Specifically, the electron-only device was designed and fabricated using ITO/Al/ETL/Al structure, as shown in the inset in Fig. 1f. In this

analysis, we assumed that the current is only related to electrons. When the effects of diffusion and the electric field are neglected, the current density can be determined by the SCLC[73]. The thickness of 80-nm Al was deposited on ITO substrates by thermal evaporation, and then the different ETLs were spin-coated on ITO/Al. Finally, 80-nm-thick Al was deposited on ITO/Al/ETL samples. The dark $J$–$V$ curves of the devices were performed on a Keithley 2400 source at ambient conditions. The electron mobility ($\mu_e$) is extracted by fitting the $J$–$V$ curves using the Mott–Gurney law (3)

$$\mu_e = \frac{8JL^3}{9\varepsilon_0\varepsilon\left(V_{app} - V_r - V_{bi}\right)^2} \qquad (3)$$

where $J$ is the current density, $L$ the thickness of different ETLs, $\varepsilon_0$ the vacuum permittivity, $\varepsilon_r$ the dielectric permittivity of various ETLs, $V_{app}$ the applied voltage, $V_r$ the voltage loss due to radiative recombination, and $V_{bi}$ the built-in voltage owing to the different work function between the anode and cathode.

**Fabrication of solar cells.** The perovskite absorbers were deposited on different ETL substrates using one-step solution processed. In total, 240.8 mg of FAI, 646.8 mg of $PbI_2$, and 18.2 mg of CsI were dissolved in 1 mL of DMF and DMSO (4:1, volume/volume), with stirring at 60 °C for 2 h. The precursor solution was spin-coated on the EDTA, $SnO_2$ and E-$SnO_2$ substrates. The spin-coated process was divided by a consecutive two-step process, the spin rate of the first step is 1000 rpm for 15 s with accelerated speed of 200 rpm, and the spin rate of the second step is 4000 rpm for 45 s with accelerated speed of 1000 rpm. During the second step end of 15 s, 200 μL of chlorobenzene was drop-coated to treat the perovskite films, and then the perovskite films were annealed at 100 °C for 30 min in a glovebox. After cooling down to room temperature, the spiro-OMeTAD solution was coated on perovskite films at 5000 rpm for 30 s with accelerated speed of 3000 rpm. The 1-mL HTL chlorobenzene solution contains 90 mg of spiro-OMeTAD, 36 μL of 4-tert-butylpyridine, and 22 μL of lithium bis(trifluoromethylsulfonyl) imide of 520 mg mL$^{-1}$ in acetonitrile. The samples were retained in a desiccator overnight to oxidate the spiro-OMeTAD. Finally, 100-nm-thick Au was deposited using thermal evaporation. The device area of 0.1134 cm$^2$ was determined by a metal mask.

**Characterization.** The $J$–$V$ curves of the PSCs were measured using a Keithley 2400 source under ambient conditions at room temperature. The light source was a 450-W xenon lamp (Oriel solar simulator) with a Schott K113 Tempax sunlight filter (Praezisions Glas & Optik GmbH) to match AM1.5 G. The light intensity was 100 mW cm$^{-2}$ calibrated by a NREL-traceable KG5-filtered silicon reference cell. The active area of 0.1017 cm$^2$ was defined by a black metal aperture to avoid light scattering into the device, and the aperture area was determined by the MICRO VUE sol 161 instrument. The $J$–$V$ curves for PSCs were tested both at reverse scan (from 2 to −0.1 V, step 0.02 V) and forward scan (from −0.1 to 2 V, step 0.02 V), and the scan rate was selected from 0.01 to 0.5 V s$^{-1}$. There was no preconditioning before the test. The IPCE was implemented on the QTest Station 2000ADI system (Crowntech. Inc., USA). AFM height images were attained by a Bruker Multimode 8 in tapping mode. KPFM was carried out on Bruker Metrology Nanoscope VIII AFM in an ambient atmosphere. The TRPL spectra were performed on an Edinburgh Instruments FLS920 fluorescence spectrometer. SEM images were gained by a field-emission scanning electron microscope (SU8020) under an accelerating voltage of 2 kV. XPS measurements were performed on an AXISULTRA X-ray photoelectron spectrometer. The optical transmission was acquired by a Hitachi U-3900 spectrophotometer.

**Data availability.** The data that support the findings of this study are available from the corresponding author upon reasonable request.

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

## Acknowledgements

The authors acknowledge support from the National Key Research Program of China (2016YFA0202403), the National Natural Science Foundation of China (61604090/91733301), the financial support from the Institute of Critical Technology and Applied Science (ICTAS), and the Shaanxi Technical Innovation Guidance Project (2018HJCG-17). S.P. would like to acknowledge the financial support from the Air Force Office of Scientific Research (A. Sayir). S.L. would like to acknowledge the support from the National University Research Fund (GK261001009), the Innovative Research Team (IRT_14R33), the 111 Project (B14041), and the Chinese National 1000-Talent-Plan program.

## Author contributions

D.Y. designed and conducted the experiments, fabricated and characterized the devices, and analyzed the data. R.Y., K.W., C.W., X.Z., and J.F. contributed to useful comments for the paper. X.R. preformed the FTIR. D.Y. wrote the first draft of the paper. S.(F.)L. and S.P. supervised the overall project, discussed the results, and contributed to the final manuscript.

## Additional information

**Competing interests:** The authors declare no competing interests.

