## [Peer Review File · Nature Communications]

Reviewers' Comments:

Reviewer #1:

Remarks to the Author:

The paper by Yang et al. describing the preparation of EDTA-complexed SnO₂ represents an interesting and obviously very high-performance perovskite solar cell electron transport layer. However, due to numerous issues with the paper, detailed below, I recommend that this paper be rejected from Nature Communications pending major revisions by the authors.

First some general comments.

- The authors consistently use hyperbolic language in this manuscript. They describe – even in the title – the solar cells as “without hysteresis” and say that hysteresis is “completely eliminated” in their devices. I agree that the devices have very low hysteresis for perovskite solar cells. This is clearly shown in Fig. S11. Nevertheless, the hysteresis is not eliminated: eliminated is simply too strong of a word. There are differences in the forward and reverse JV scans. While these differences are slight they are clearly visible. Fig. 6 shows the FF changes from 0.792 to 0.783 with scan direction. This is hysteresis. The authors should modify the language of the paper in this respect. Hysteresis is reduced, it is perhaps even negligible, but it is certainly still present and is not “eliminated”.
- The English throughout the paper requires significant improvement to improve the readability of the manuscript. This is particularly true of the methods section. In fact, it is in places difficult to understand the methods being used due to inconsistent grammar. Improving readability would greatly improve the impact of this work.
- The authors use a variety of device configurations to measure material properties such as defect density, mobility, and others. While I don't necessarily disagree or refute the results as presented, there are complicating factors associated with these measurements and device architectures. The authors should be clear about the caveats associated with their measurements and why they believe these can be neglected in this case. This will give the reader a fairer picture of the measurements undertaken and the conclusions of the work.

More specific comments.

- The authors' description of the crystallization with respect to the Gibbs free energy is unclear. What do the authors mean that the “hydrophilicity of the E-SnO₂ surface... decreases the Gibbs barrier during perovskite growth” and how do the authors envision this resulting in “improved quality of perovskite film”? What Gibbs barrier is being reduced? Are the authors merely expecting more uniform nucleation of perovskite crystals on the E-SnO₂ surface? It isn't clear from the discussion of this in the text what exactly is meant and how this would affect trap densities in the perovskite layer. Also, it looks like these N_t measurements are conducted with FAPbI₃. Is this a different active layer than the FACsPbI₃ layers used in the rest of this work? If so, why?
- The authors state that hysteresis is more severe in planar-type architectures (see pg 3). This is not universally a true statement. Generally p-i-n structures have very low hysteresis. Moreover, Jiang et al. published a report in Nature Energy which demonstrated that the control SnO₂ ETL used in the present work can have very low hysteresis (10.1038/nenergy.2016.177).
- On page 4 the authors comment on the high annealing temperature necessary for TiO₂. Contrary to this assertion, TiO₂ ETLs which can be processed at low temperature with high performance have been widely shown. Including some with similar high performance to the current study. See DOI: 10.1126/science.aai9081
- The E-SnO₂ solution is said to be stable. Can the authors provide device data or something else to actually confirm this? A photograph of the solution does not confirm if the solution is actually stable or not.
- The authors say they obtain their XPS data from films on quartz substrates. What effects from charging do the authors see with this? I would expect very significant charging in these samples which could greatly shift the observed binding energies.
- The FTIR spectra of the control SnO₂ is missing. The authors show the O-O stretch at ~1000 cm⁻¹, is this related to the SnO₂ itself exposed to O₂ (as described in the reference cited) or is it a result of the EDTA bound to the SnO₂ surface?
- The PL lifetimes in Fig 3c are very fast! lifetimes of less than 1 ns for good perovskite films

indicate either something really strange going on or nonlinear behavior. The assignment to interface quenching and trap recombination is complicated by this. Is the lifetime excitation intensity dependent? Typical lifetimes are more in the 50-1000 ns range. Related to this, how much does ITO quench the emission?

- On page 17 the authors state that the “electron mobility of E-SnO₂ would enhance electron transfer,” do the authors mean transport?
- The authors write dimethyl sulfide in the materials and methods, is this a typo of dimethyl sulfoxide or is this correct?
- Is the SnO₂/water solution 2.5 wt%?
- Are the SnO₂ layers annealed at all?
- How are the control SnO₂ layers and the control EDTA layers fabricated?

Reviewer #2:

Remarks to the Author:

The authors report a success strategy in eliminating hysteresis and at the same time attaining record-efficiency in planar-type perovskite solar cells by using EDTA-complexed SnO₂ (E-SnO₂) electron transport layers (ETL). Statistical analysis seems sound as well as experiments carried out for improving understanding. I thus recommend publication after addressing these minor but useful points below.

“Even though ETL free planar-type PSCs have been reported, their performances are poor compared to those with ETL”: Please quantify the PCE of these ETL-free devices.

“ii) suitable energy level with the perovskite materials to reduce the energy barrier for electron transport” On one hand you want to eliminate an energy barrier for electron injection, on the other you don’t want the conduction band to be too much lower than that of the perovskite otherwise it would lower the Voc. Please explain this better.

“However, the PCE of the planar-type PSCs is still lower than that of the mesoporous-type devices because there exists significant energy barrier between SnO₂ ETL and perovskite absorber, leading to energy loss.” In fact some referenced have used both SnO₂ compact layer and either a compact layer (<https://www.sciencedirect.com/science/article/pii/S092702481730065X>) or a mesoporous TiO₂ layer (<https://link.springer.com/article/10.1007/s12274-017-1896-5>) over the top to improve the efficiency confirming what the authors say. Other groups have doped the SnO₂ for improved performance (<https://pdfs.semanticscholar.org/cf6b/9a40dcd8a82887d20b23354697dd873e7b4.pdf>). I would recommend adding this further discussion and references in the introduction for the state of the art.

On page 7 “The reduced energy barrier is also believed to enhance charge transfer from the perovskite to the ETL” . This is not an energy barrier because the conduction band of the SnO₂ is lower than that of the perovskite. The electron in the solar cells is injected from the perovskite into the SnO₂ during solar cell operation (this is different from IV curves in the dark where electrons are injected from the SnO₂ into the perovskite). Why should the transfer be better if the jump is lower? The Voc should be higher because the conduction band of the SnO₂ is closer to that of the perovskite. Please review this explanation as well as the caption of S5.

The authors partly explain the higher crystallinity of the perovskite films using contact angle measurements, where larger grain sizes are correlated with lower contact angles. However, there is literature where investigators say that non wetting surfaces lead to higher grain size. See <https://www.nature.com/articles/ncomms8747>

Although others do indeed correlate crystallinity with hydrophobicity

<https://www.ncbi.nlm.nih.gov/pubmed/27760287>

<https://nanoscalereslett.springeropen.com/articles/10.1186/s11671-016-1540-4>

<https://link.springer.com/article/10.3938/jkps.69.406>

Can the authors discuss this matter in more depth using the literature and provide a clearer or

more definitive explanation of the literature on this matter?

Again on page 11 when discussing why photoluminescence times are shorter, the explanation of higher mobility seems correct, but the reduced energy barrier does not (it is a jump). It may be that the interface or adhesion is better at the interface. Same when discussing hysteresis at the end of page 16 "meaning no energy barrier for electron transfer (Fig. S5), that is expected to facilitate electron extraction from the perovskite to the ETL." The explanation must be another. Maybe less traps? Better initial growth on the E-SnO₂?

On page 17 the authors use a bending radius of 7mm. Why was this chosen? It is where ITO cracks? Please provide some references.

Reviewer #3:

Remarks to the Author:

In this work, the authors introduced an ethylene diamine tetraacetic acid (EDTA)-complexed SnO₂ as the electron transport layer (ETL) for planar perovskite solar cells (PSCs) to realize a certified efficiency of 21.5% with eliminated hysteresis and enhanced stability. However, similar concept has been already reported in a previous work (Chem. Mater., 2017, 29, 4176–4180) besides the improved efficiency for PSCs. I thus felt the novelty of this work is not impressive and not suitable to publish in Nature Communications as considering its high standard. I would recommend the publication of this work in the other specific journal.

Some remarks / questions follow:

1. The author should refer the mentioned reference (Chem. Mater., 2017, 29, 4176–4180) in the manuscript.
2. The surface potential (Fermi level) obtained from KPFM is totally different from the value of CB band, and the energy diagram in figure 1(d) is misleading.
3. What is the real mechanism for the eliminated hysteresis by using EDTA-SnO₂? The author should clarify it.
4. As known, in the conventional structure of PSC, the instability is mainly due to the perovskite layer and spiro-OMeTAD HTL. As the author did not change the perovskite layer and HTL, why the device without encapsulation could show better stability in air with 35% humidity for a so long time of 3000 h? The author should clarify it.

Point-By-Point Response to Referees' Comments

Reviewer #1 (Remarks to the Author):

Introductory comment:

The paper by Yang et al. describing the preparation of EDTA-complexed SnO₂ represents an interesting and obviously very high-performance perovskite solar cell electron transport layer. However, due to numerous issues with the paper, detailed below, I recommend that this paper be rejected from Nature Communications pending major revisions by the authors.

Response to introductory comment: Thank you for your insightful comments and suggestions! We have conducted additional experiments and revised the manuscript to address all the comments and questions raised in the review.

Comments:

First some general comments.

- The authors consistently use hyperbolic language in this manuscript. They describe – even in the title-the solar cells as “without hysteresis” and say that hysteresis is “completely eliminated” in their devices. I agree that the devices have very low hysteresis for perovskite solar cells. This is clearly shown in Fig. S11. Nevertheless, the hysteresis is not eliminated: eliminated is simply too strong of a word. There are differences in the forward and reverse JV scans. While these differences are slight they are clearly visible. Fig. 6 shows the FF changes from 0.792 to 0.783 with scan direction. This is hysteresis. The authors should modify the language of the paper in this respect. Hysteresis is reduced, it is perhaps even negligible, but it is certainly still present and is not “eliminated”.*

Response: Thank you very much for your suggestion. We agree with your comment and have revised all relevant sections accordingly.

- *The English throughout the paper requires significant improvement to improve the readability of the manuscript. This is particularly true of the methods section. In fact, it is in places difficult to understand the methods being used due to inconsistent grammar. Improving readability would greatly improve the impact of this work.*

Response: Thanks for the suggestion! Revised version has been reviewed by a native English speaker for grammar.

- *The authors use a variety of device configurations to measure material properties such as defect density, mobility, and others. While I don't necessarily disagree or refute the results as presented, there are complicating factors associated with these measurements and device architectures. The authors should be clear about the caveats associated with their measurements and why they believe these can be neglected in this case. This will give the reader a fairer picture of the measurements undertaken and the conclusions of the work.*

Response: Thank you very much for the comments! Per your request, we have revised the sections with proper analyses and discussion, as shown on pages 10 and 22.

To gain insights into the charge transport, we have measured electron mobility using different electron transport layers (ETLs) in the same device structure. Specifically, the electron-only device was designed and fabricated using structure ITO/Al/ETL/Al, as shown in the inset in Fig. 1f. As is well-described in literature (ACS Energy Lett., 2017, 2, 2667-2673; Joule, 2018, 2, 168-183; Adv. Sci., 2016, 3, 1600027, etc.), the space charge limited current (SCLC) model was used to estimate the electron mobility. In this analysis, we assumed that the current is only

related to electrons. When the effects of diffusion and the electric field are neglected, the current density can be determined by the SCLC.^{R2} (Page 22).

Likewise, the trap state density (N_t) of the perovskite films are estimated using the device structure ITO/ETL/perovskite/PCBM/Ag, similar to previously reported results (Energy Environ. Sci., 2017, 10, 2570-2578; Adv. Mater., 2016, 28, 5206-5213; Energy Environ. Sci., 2017, 10, 2095-2102, etc.),^{R3-R5} The device architecture is shown in Fig. 3a with the dark condition I-V curves. The linear correlation in the I-V curve reveals an ohmic-type device response in the low bias voltage range. The current is quickly increased nonlinearly when the bias voltage exceeds the trap-filled limit voltage (V_{TFL}) located at the kink point, indicating that the traps are completely filled. Hence, the trap density of the perovskite film can be calculated using V_{TFL} through equation (2) presented in the manuscript. More details are provided in the manuscript on page 10.

More specific comments.

- *The authors' description of the crystallization with respect to the Gibbs free energy is unclear. What do the authors mean that the "hydrophilicity of the E-SnO₂ surface... decreases the Gibbs barrier during perovskite growth" and how do the authors envision this resulting in "improved quality of perovskite film"? What Gibbs barrier is being reduced? Are the authors merely expecting more uniform nucleation of perovskite crystals on the E-SnO₂ surface? It isn't clear from the discussion of this in the text what exactly is meant and how this would affect trap densities in the perovskite layer. Also, it looks like these N_t measurements are conducted with FAPbI₃. Is this a different active layer than the FACsPbI₃ layers used in the rest of this work? If so, why?*

Response: According to the established model for nucleation and growth of thin films,^{R6,R7} the perovskite formation process can be divided into four steps: i) formation of crystal nucleus, ii) evolution of nuclei's into island structure, iii) formation of networked microstructure and iv) growth of networks into continuous film. The Gibbs free energy for heterogeneous nucleation in the first step can be expressed as:

$$\Delta G_{\text{heterogeneous}} = \Delta G_{\text{homogeneous}} \times f(\theta),$$

wherein $f(\theta) = (2 - 3\cos\theta + \cos^3\theta)/4$, and θ is contact angle of the precursor solution. Since the magnitude of θ varies in the range of $[0, \pi/2]$, the larger the is θ the smaller is the magnitude of $\cos\theta$, and therefore larger is the parameter $f(\theta) \in [0, 1]$. In other words, a smaller contact angle results in reduced Gibbs free energy for heterogeneous nucleation, thereby, assisting the nucleation process. Higher nucleation density will promote the film densification process.^{R7} Compared to EDTA and SnO₂, E-SnO₂ shows the smallest contact angle (20.67°, Fig. S6), which is further expected to improve the wettability between the E-SnO₂ and the perovskite layer.^{R8-R10} Thus, the perovskite coated on the E-SnO₂ exhibits better crystallinity (Fig. S7) and full surface coverage (Fig. 2c). In addition, the small contact angle of substrate provides the low surface energy,^{R11} leading to increased grain size during the growth of the networked structure as observed in the SEM measurements.^{R7} This explanation has been added in the manuscript (page 9-10).

The N_t measurements are conducted with FA_{0.95}CS_{0.05}PbI₃, the same active layer used in the rest of this work. We are really sorry for the error label in Fig. 3a. It has been corrected.

• *The authors state that hysteresis is more severe in planar-type architectures (see pg 3). This is not universally a true statement. Generally p-i-n structures have very low hysteresis. Moreover,*

Jiang et al. published a report in Nature Energy which demonstrated that the control SnO₂ ETL used in the present work can have very low hysteresis (10.1038/nenergy.2016.177).

Response: Thank you very much for your valuable comments! We have revised the statement accordingly (page 3-4) as follows: “Generally, the inverted device structure utilizing fullerene ETL display very low hysteresis, however, it usually yields lower PCE, not to mention that fullerene is very expensive.^{26,27}” “Recently, Jiang *et al.* developed the SnO₂ nanoparticles as the ETL and demonstrated certified efficiency as high as 19.9% with very low hysteresis.²¹ However, the PCE of the planar-type PSCs is still lower than that of the mesoporous-type devices likely due to charge accumulation at the ETL/perovskite interface caused by relatively low electron mobility of the ETL.⁴⁴ It is expected that better PSC performance will be achieved by increasing electron mobility of the ETLs.”

• *On page 4 the authors comment on the high annealing temperature necessary for TiO₂. Contrary to this assertion, TiO₂ ETLs which can be processed at low temperature with high performance have been widely shown. Including some with similar high performance to the current study. See DOI: 10.1126/science.aai9081*

Response: Thank you for your valuable comments. Recent studies have demonstrated that TiO₂ with good properties can be fabricated at low temperature using techniques such as magnetron sputtering, surface modification, etc.

Per your suggestion, we have revised the relevant description in the manuscript (page 4) as follows: “There have been extensive efforts in developing low temperature TiO₂ ETL, such as exploring low temperature synthesis processes through doping and chemical engineering. Results

shown by Tan *et al.* demonstrate that use of chlorine to modify the TiO₂ microstructure at low temperatures provides promising PCE of 20.1%.³⁵”

- The E-SnO₂ solution is said to be stable. Can the authors provide device data or something else to actually confirm this? A photograph of the solution does not confirm if the solution is actually stable or not.

Response: Thank you for your question! Fig. S2 compares the FTIR spectra of the E-SnO₂ solution measured in the freshly prepared condition and again after it was stored in ambient for 2 months. It is clear that there is no obvious difference between the two solutions indicating the high stability. We have added this measurement in the Supplementary Information (Fig. S2).

Figure S2. The FTIR and photographs of the E-SnO₂ solution taken under fresh condition and after it was stored in ambient for 60 days.

- The authors say they obtain their XPS data from films on quartz substrates. What effects from charging do the authors see with this? I would expect very significant charging in these samples which could greatly shift the observed binding energies.

Response: The XPS of EDTA, SnO₂, and E-SnO₂ films deposited on quartz substrates were measured again. In order to reduce the charging effect, the exposed surface of the quartz substrate was coated with a conductive silver paint and connected to the ground. We calibrated the binding energy scale for all XPS measurements to the carbon 1s line at 284.8 eV. It is clear from these measurements that SnO₂ shows only peaks attributed to Sn and O. After the EDTA treatment, the E-SnO₂ film shows an additional peak located at ca. 400 eV, ascribed to N. Meanwhile, the Sn 3d peaks from E-SnO₂ are shifted by ca. 0.16 eV in contrast to the pristine SnO₂ (Fig. S3), indicating that EDTA is bound to the SnO₂. These results have been described in the manuscript (page 6, Fig. 1a) and the Supplementary Information (Fig. S3).

• *The FTIR spectra of the control SnO₂ is missing. The authors show the O-O stretch at ~1000 cm⁻¹, is this related to the SnO₂ itself exposed to O₂ (as described in the reference cited) or is it a result of the EDTA bound to the SnO₂ surface?*

Response: We have measured the FTIR spectra again from 3600 cm⁻¹ to 500 cm⁻¹ for all three samples EDTA, SnO₂ and E-SnO₂. As shown in Fig. 1b, the peaks around 2895 cm⁻¹ and 1673 cm⁻¹ belong to C-H and C=O stretching vibration in the EDTA, respectively. The characteristic peaks of SnO₂ observed at ca. 701 cm⁻¹ and 549 cm⁻¹ are due to O-Sn-O stretch and the Sn-O vibration, respectively.^{R12} In addition, the peak at 1040 cm⁻¹ in the SnO₂ film is attributed to O-O stretching vibration due to oxygen adsorption on the SnO₂ surface.^{R13} For the E-SnO₂ sample, the characteristic peaks of SnO₂ shift to 713 cm⁻¹ and 563 cm⁻¹, and the C-H and C=O stretching vibration peaks shift to 2913 cm⁻¹ and 1624 cm⁻¹, further demonstrating that the EDTA is indeed complexed with SnO₂. We have added this additional analysis in the manuscript (page 6, Fig. 1b).

• The PL lifetimes in Fig 3c are very fast! lifetimes of less than 1 ns for good perovskite films indicate either something really strange going on or nonlinear behavior. The assignment to interface quenching and trap recombination is complicated by this. Is the lifetime excitation intensity dependent? Typical lifetimes are more in the 50-1000 ns range. Related to this, how much does ITO quench the emission?

Response: Thank you for your insightful question. We measured the lifetime of the perovskite films deposited on various substrates using different excitation intensity. The TRPL spectra and fitting data are shown in Fig. S8 and Table S1. Generally, the slow decay component (τ_1) is attributed to the radiative recombination of free charge carriers due to traps in the bulk, and the fast decay component (τ_2) is originated from the quenching of charge carriers at interface.^{R14} The glass/perovskite sample shows the longest lifetime under excitation intensity of 3 $\mu\text{J}/\text{cm}^2$. For perovskite coated on the ITO substrate, the lifetime is decreased more than half due to the charge transfer from perovskite into ITO. For EDTA/perovskite and SnO_2 /perovskite samples, both the fast and slow decay lifetimes are very similar, and τ_1 dominates the PL decay for both samples, indicating severe recombination before they were extracted. When the perovskite is deposited on E- SnO_2 , both τ_1 and τ_2 were shortened to 14.16 ns and 0.97 ns, with a proportion of 45.32% and 54.68%, respectively. Meanwhile, τ_2 appears to dominate the PL decay, indicating that electrons are effectively extracted from the perovskite layer to the E- SnO_2 with minimal recombination loss. Even under smaller excitation intensity (0.5 $\mu\text{J}/\text{cm}^2$), the acceleration of the lifetime for E- SnO_2 /perovskite is observed. The lifetime increases with reduced excitation intensity (Fig. S8 and Table S1), in agreement with previous report.^{R15} The electron transport yield (Φ_{tr}) of different ETLs with different excitation intensity can be estimated using equation, $\Phi_{\text{tr}} = 1 - \tau_{\text{p}}/\tau_{\text{glass}}$, where τ_{p} is the average lifetime for perovskite deposited on different substrates, and

τ_{glass} is the average lifetime for glass/perovskite. With the excitation intensity of $3 \mu\text{J}/\text{cm}^2$, the electron transport yields of ITO, EDTA, SnO_2 and E- SnO_2 are 49.72%, 67.58%, 68.31% and 81.50%, respectively. When the excitation intensity reduces to $0.5 \mu\text{J}/\text{cm}^2$, the electron transport yields of ITO, EDTA, SnO_2 and E- SnO_2 are increased to 60.37%, 74.46%, 80.65% and 90.82%, respectively. It is clear that the excitation intensity can significantly increase the electron transport yield. These results further indicate that the E- SnO_2 is a good electron extraction layer for planar-type perovskite solar cells (PSCs). We have added the details in the manuscript (page 12) and the Supplementary Information (Fig. S8 and Table S1).

Figure S8. TRPL spectra of perovskite films deposited on different substrates using excitation intensity of (a) $3 \mu\text{J}/\text{cm}^2$ and (b) $0.5 \mu\text{J}/\text{cm}^2$.

Table S1 | Parameters of the TRPL spectra of perovskite films deposited on different substrates under various excitation intensity.

Excitation intensity	Sample	τ_{ave} (ns)	τ_1 (ns)	% of τ_1	τ_2 (ns)	% of τ_2
$3 \mu\text{J}/\text{cm}^2$	Glass/perovskite	71.07	76.27	72.35	20.56	27.65
	ITO/perovskite	35.73	36.37	68.24	1.46	31.76
	ITO/EDTA/perovskite	23.04	23.74	65.29	1.44	34.71
	ITO/ SnO_2 /perovskite	22.52	23.26	57.82	1.09	42.18

	ITO/E-SnO ₂ /perovskite	13.15	14.16	45.32	0.97	54.68
	Glass/perovskite	100.30	105.33	76.72	22.38	23.28
	ITO/perovskite	58.50	65.36	63.33	20.27	36.67
0.5 μJ/cm ²	ITO/EDTA/perovskite	37.71	42.73	60.25	13.35	39.75
	ITO/SnO ₂ /perovskite	28.57	33.33	52.16	8.71	47.84
	ITO/E-SnO ₂ /perovskite	13.55	16.40	49.17	6.02	50.83

• On page 17 the authors state that the “electron mobility of E-SnO₂ would enhance electron transfer,” do the authors mean transport?

Response: We are sorry for the typo. This sentence should have been: “the high electron mobility of E-SnO₂ would enhance electron transport from perovskite to E-SnO₂ ETL,” We have made the correction in revised manuscript (page 18).

• The authors write dimethyl sulfide in the materials and methods, is this a typo of dimethyl sulfoxide or is this correct?

Response: Thank you very much for the catch. This is a typo, and we have corrected it as “dimethyl sulfoxide” in the manuscript.

• Is the SnO₂/water solution 2.5 wt%?

Response: Yes, this is the weight concentration. The SnO₂ aqueous colloidal dispersion (15 wt%, purchased from Alfa Aesar) was diluted using deionized water to the concentration of 2.5 wt%. Details are presented in the Methods Section (page 21).

• Are the SnO₂ layers annealed at all?

Response: The ETLs including EDTA, SnO₂ and E-SnO₂ were kept at 60 °C in a vacuum oven after spin-coating. The system was evacuated to ca. 5 Pa for 30 min to remove residual solvent, as described in the Methods Section (page 21).

• *How are the control SnO₂ layers and the control EDTA layers fabricated?*

Response: We have added details on fabrication process for the SnO₂ and the EDTA layers in the Methods Section (page 21): The 0.2 mg EDTA was dissolved in 1 mL deionized water, and the SnO₂ aqueous colloidal dispersion (15 wt%) was diluted using deionized water to the concentration of 2.5 wt%. These solutions were stirred at room temperature for 2 hours. The SnO₂ and EDTA layers were fabricated by spin-coating at 5000 rpm for 60 s using the corresponding solution, and then dried in a vacuum oven at 60 °C under ca. 5 Pa for 30 min to remove residual solvent.

Reviewer #2 (Remarks to the Author):

Introductory comment:

The authors report a success strategy in eliminating hysteresis and at the same time attaining record-efficiency in planar-type perovskite solar cells by using EDTA-complexed SnO₂ (E-SnO₂) electron transport layers (ETL). Statistical analysis seems sound as well as experiments carried out for improving understanding. I thus recommend publication after addressing these minor but useful points below.

Response to Introductory comment: Thank you very much for the nice comments.

Comments:

“Even though ETL free planar-type PSCs have been reported, their performances are poor compared to those with ETL”: Please quantify the PCE of these ETL-free devices.

Response: We have briefly summarized the PCE of ETL-free PSCs in the manuscript: “Even though ETL free planar-type PSCs have been reported,^{30,31} their highest PCE is only 14.14%, significantly lower than that of the cells with ETL, demonstrating the importance of the ETL in this configuration of PSCs.” We have added these comments in the manuscript (page 3).

ii) suitable energy level with the perovskite materials to reduce the energy barrier for electron transport” On one hand you want to eliminate an energy barrier for electron injection, on the other you don’t want the conduction band to be too much lower than that of the perovskite otherwise it would lower the V_{oc} . Please explain this better.

Response: Thanks for the comment and excellent question. The PSCs using the E-SnO₂ ETL show higher J_{sc} due to the high electron mobility of E-SnO₂. Further, they show larger V_{oc} due to the energy level match between the Fermi level of E-SnO₂ and the conduction band of the perovskite. This information has been added in the manuscript (page 3).

“However, the PCE of the planar-type PSCs is still lower than that of the mesoporous-type devices because there exists significant energy barrier between SnO₂ ETL and perovskite absorber, leading to energy loss.” In fact some referenced have used both SnO₂ compact layer and either a compact layer (<https://www.sciencedirect.com/science/article/pii/S092702481730065X>) or a mesoporous TiO₂ layer (<https://link.springer.com/article/10.1007/s12274-017-1896-5>) over the top to improve the efficiency confirming what the authors say. Other groups have doped the SnO₂ for improved performance (<https://pdfs.semanticscholar.org/cf6b/9a40dcdb8a82887d20b23354697dd873e7b4.pdf>). I would

recommend adding this further discussion and references in the introduction for the state of the art.

Response: Thanks for these suggestions. We have incorporated this information in the introduction (page 4): “The SnO₂-TiO₂ (planar and mesoporous) composite layers were developed to enhance the performance of the PSCs.^{37,38} It is noteworthy to mention that Al³⁺-doped SnO₂ provides even better performance.³⁹”. All these literatures have been cited as No. 37-39 in the Reference Section.

On page 7 “The reduced energy barrier is also believed to enhance charge transfer from the perovskite to the ETL”. This is not an energy barrier because the conduction band of the SnO₂ is lower than that of the perovskite. The electron in the solar cells is injected from the perovskite into the SnO₂ during solar cell operation (this is different from I-V curves in the dark where electrons are injected from the SnO₂ into the perovskite). Why should the transfer be better if the jump is lower? The V_{oc} should be higher because the conduction band of the SnO₂ is closer to that of the perovskite. Please review this explanation as well as the caption of S5.

Response: Thank you very much for your suggestion. The effective electron injection is attributed to the high electron mobility of E-SnO₂ while the larger V_{oc} (Fig. 4a and Table 1) results from the smaller energy level difference between E-SnO₂ and the perovskite layer. We have revised the explanation in the manuscript (page 8) as “The Fermi level of E-SnO₂ is very close to the conduction band of perovskite, which is beneficial for enhancing V_{oc}.”

In addition, we have revised the caption of Fig. S5 as “In order to examine electron transport capability of the E-SnO₂ film, the glass/ITO/ETL/perovskite/PCBM/Al devices were fabricated. When a voltage is applied to the top ITO electrode, electrons are injected to ETL

from the perovskite. The J - V curve of the E-SnO₂-based device exhibits lower response voltage than that of the SnO₂-based device (Fig. S5a), indicating that the electron injection from the perovskite to the E-SnO₂ is easier than to the SnO₂, due to the higher electron mobility of E-SnO₂. In addition, Fig. S5b and S5c illustrate the energy level alignment for the devices based on the SnO₂ and E-SnO₂ ETLs. It is clear that the Fermi level of the E-SnO₂ shows a better match to the conduction band of the perovskite than that obtained for the SnO₂. This provides enhancement in the observed V_{oc} (Fig. 4a and Table 1).”

The authors partly explain the higher crystallinity of the perovskite films using contact angle measurements, where larger grain sizes are correlated with lower contact angles. However, there is literature where investigators say that non wetting surfaces lead to higher grain size. See <https://www.nature.com/articles/ncomms8747>

Although others do indeed correlate crystallinity with hydrophilicity

<https://www.ncbi.nlm.nih.gov/pubmed/27760287>

<https://nanoscalereslett.springeropen.com/articles/10.1186/s11671-016-1540-4>

<https://link.springer.com/article/10.3938/jkps.69.406>

Can the authors discuss this matter in more depth using the literature and provide a clearer or more definitive explanation of the literature on this matter?

Response: Thank you very much for your suggestions! According to the established model for nucleation and growth of thin films,^{R6,R7} the perovskite formation process can be divided into four steps: i) formation of crystal nucleus, ii) evolution of nuclei's into island structure, iii) formation of networked microstructure and iv) growth of networks into continuous film. The Gibbs free energy for heterogeneous nucleation in the first step can be expressed as:

$$\Delta G_{\text{heterogeneous}} = \Delta G_{\text{homogeneous}} \times f(\theta),$$

wherein $f(\theta) = (2 - 3\cos\theta + \cos^3\theta)/4$, and θ is contact angle of the precursor solution. Since the magnitude of θ varies in the range of $[0, \pi/2]$, the larger the is θ the smaller is the magnitude of $\cos\theta$, and therefore larger is the parameter $f(\theta) \in [0, 1]$. In other words, a smaller contact angle results in reduced Gibbs free energy for heterogeneous nucleation, thereby, assisting the nucleation process. Higher nucleation density will promote the film densification process.^{R7} Compared to EDTA and SnO₂, E-SnO₂ shows the smallest contact angle (20.67°, Fig. S6), which is further expected to improve the wettability between the E-SnO₂ and the perovskite layer.^{R8-R10} Thus, the perovskite coated on the E-SnO₂ exhibits better crystallinity (Fig. S7) and full surface coverage (Fig. 2c). In addition, the small contact angle of substrate provides the low surface energy,^{R11} leading to increased grain size during the growth of the networked structure as observed in the SEM measurements.^{R7} This explanation has been added in the manuscript (page 9-10), and the relevant literatures have been cited in the Reference Section.

Again on page 11 when discussing why photoluminescence times are shorter, the explanation of higher mobility seems correct, but the reduced energy barrier does not (it is a jump). It may be that the interface or adhesion is better at the interface. Same when discussing hysteresis at the end of page 16 “meaning no energy barrier for electron transfer (Fig. S5), that is expected to facilitate electron extraction from the perovskite to the ETL.” The explanation must be another. Maybe less traps? Better initial growth on the E-SnO₂?

Response: Thank you very much for your suggestions. The perovskite deposited on E-SnO₂ substrate shows the shortest lifetime (Fig. 3c and Table S1), which is attributed to its high electron mobility. The short lifetime indicates that the carriers can be effectively extracted into

E-SnO₂ with minimal recombination loss. We have revised this explanation in the manuscript (page 12).

Generally, the hysteresis of PSCs is ascribed to ion migration, high trap density, and unbalanced charge transport within the perovskite device. The trap density of the perovskite film is significantly reduced when deposited on the E-SnO₂, which is one reason for reduced hysteresis in the PSCs. In addition, the electron mobility of E-SnO₂ ETL is $2.27 \times 10^{-3} \text{ cm}^2 \text{ V}^{-1} \text{ s}^{-1}$ (Fig. 1f), comparable to the hole mobility of the doped spiro-OMeTAD ($\sim 10^{-3} \text{ cm}^2 \text{ V}^{-1} \text{ s}^{-1}$) HTL. Thus, the electron flux (F_e) is essentially equal to the hole flux (F_h) because the interface area of the ETL/perovskite is the same as that of the perovskite/HTL (Fig. S13b). Therefore, there is no significant charge accumulation, and consequently, the devices based on the E-SnO₂ exhibit negligible hysteresis. We have added these explanations in the manuscript and the Supplementary Information (page 18, Fig. S13).

On page 17 the authors use a bending radius of 7mm. Why was this chosen? It is where ITO cracks? Please provide some references.

Response: According to a previous report,^{R16} it is safe for ITO to be bended to a radius of 14 mm, and when the bending radius is smaller than 14 mm, the ITO layer starts to crack, leading to significant degradation in conductivity. In order to examine the intrinsic mechanical stability of the flexible PSCs, we therefore adopted the bending radius of 7 mm, half of that suggested in literature, to test the flexible device.

Reviewer #3 (Remarks to the Author):

Introductory comment:

In this work, the authors introduced an ethylene diamine tetraacetic acid (EDTA)-complexed SnO₂ as the electron transport layer (ETL) for planar perovskite solar cells (PSCs) to realize a certified efficiency of 21.5% with eliminated hysteresis and enhanced stability. However, similar concept has been already reported in a previous work (Chem. Mater., 2017, 29, 4176-4180) besides the improved efficiency for PSCs. I thus felt the novelty of this work is not impressive and not suitable to publish in Nature Communications as considering its high standard. I would recommend the publication of this work in the other specific journal.

Response to introductory comment: We thank the Referee for helpful feedback and comments. It was pointed out that our approach is similar to that reported by Li et al. (Chem. Mater., 2017, 29, 4176-4180). In this reference, EDTA was used to modify ZnO to form a hybrid interface in organic solar cells. However, the EDTA-ZnO does not work in the PSCs because when the perovskite film is deposited onto the EDTA-ZnO surface, it shows severe degradation due to the hydroxyl groups or acetate ligands on the ZnO surface, and proton transfer reactions at the perovskite/ZnO interface.^{R17}

In our work we have used EDTA to modify the SnO₂. It is demonstrated that the efficiency of PSCs is increased to record high 21.60% (certified efficiency at 21.52% by Newport) with negligible hysteresis, the highest efficiency reported so far for the planar-type PSCs. In addition, the solar cells show significantly improved stability. We provide systematic fundamental analysis to elucidate the mechanisms responsible for this improved enhancement. Thus, we truly believe that our work advances the state-of-the-art and provides a promising pathway for obtaining high performance of planar-type PSCs. Hope the reviewer will find our explanations to his/her comments satisfactory.

Comments:

Some remarks / questions follow:

1. The author should refer the mentioned reference (Chem. Mater., 2017, 29, 4176-4180) in the manuscript.

Response: We have cited this reference as No. 45 in the manuscript (page 4-5) as: “Ethylene diamine tetraacetic acid (EDTA) provides excellent modification of ETLs in organic solar cells owing to its strong chelation function. Li et al. have employed EDTA to passivate ZnO based ETL and demonstrated improved performance of the organic solar cells.⁴⁵ However, when the EDTA-ZnO layer is used in the present perovskite cells, the hydroxyl groups or acetate ligands on the ZnO surface react with the perovskite and proton transfer reactions occur at the perovskite/ZnO interface, leading to poor device performance.^{46,}”

2. The surface potential (Fermi level) obtained from KPFM is totally different from the value of CB band, and the energy diagram in figure 1(d) is misleading.

Response: Thank you very much for the comment. The Fermi level obtained from KPFM is indeed different from the value of CB band. We are really sorry for the inappropriate label used in Fig. 1b, and it is corrected in the revised manuscript. In present work, we used the KPFM to test surface potential of different ETLs, and the images are shown in Fig. S4 in the Supplementary Information. The Fermi level (FL) of different samples can be calculated using the equation:

$$FL = 4.6 + e(SP_{\text{HOPG}} - SP_{\text{sample}}),$$

where e is the elementary charge of the electron, SP_{HOPG} and SP_{sample} are surface potential of HOPG and the sample, respectively. Thus we can obtain the values of FL for EDTA, SnO_2 and E- SnO_2 . More details are provided in the Supplementary Information (page S3, Fig. S4).

Figure 1. (d) Schematic illustration of Fermi level of EDTA, SnO_2 , and E- SnO_2 relative to the conduction band of the perovskite layer. The Fermi level of EDTA, SnO_2 , E- SnO_2 are measured by KPFM, and conduction band and valence band of the perovskite materials are obtained from the previous report (Chem. Sci., 2017, 8, 800-805).

3. What is the real mechanism for the eliminated hysteresis by using EDTA- SnO_2 ? The author should clarify it.

Response: The hysteresis of PSCs is ascribed to interfacial capacitance caused by charge accumulation at the interface, which originates from ion migration, high trap density, and unbalanced charge transport within the perovskite device.^{R18-R20} It is found that the trap density of the perovskite film is significantly reduced when it is deposited on the E- SnO_2 , one of the primary reasons for reduced hysteresis. In addition, the electron mobility of the SnO_2 ETL is only $9.92 \times 10^{-4} \text{ cm}^2 \text{ V}^{-1} \text{ s}^{-1}$ (Fig. 1f), about an order of magnitude slower than the hole mobility of the doped spiro-OMeTAD ($\sim 10^{-3} \text{ cm}^2 \text{ V}^{-1} \text{ s}^{-1}$) HTL. Thus, the electron flux (F_e) is ca. 10 times smaller than the hole flux (F_h) because the interface area of the ETL/perovskite is the same as

that of the perovskite/HTL, leading to accumulated charge or capacitance at the SnO₂/perovskite interface, as shown in Fig. S13a. The accumulated charge will cause hysteresis in the solar cells (Fig. 6c). When the high electron mobility E-SnO₂ ($2.27 \times 10^{-3} \text{ cm}^2 \text{ V}^{-1} \text{ s}^{-1}$) is employed as the ETL, the F_e is comparable to the F_h of the spiro-OMeTAD HTL (Fig. S13b), resulting in equivalent charge transport at both electrodes. Therefore, there is no significant charge accumulation or capacitance and thus reduced hysteresis in the PSCs. We have added these discussions in the manuscript and the Supplementary Information (page 18, Fig. S13).

Figure S13. Charge transport mechanism. (a) Planar-type PSCs with SnO₂ and (b) E-SnO₂.

4. As known, in the conventional structure of PSC, the instability is mainly due to the perovskite layer and spiro-OMeTAD HTL. As the author did not change the perovskite layer and HTL, why the device without encapsulation could show better stability in air with 35% humidity for a so long time of 3000 h? The author should clarify it.

Response: The instability of PSC is mainly caused by degradation of perovskite film and spiro-OMeTAD HTL. In the present work, all devices used the same spiro-OMeTAD HTL, therefore the degradation from the HTL should be same for all the devices. It is found that the grain size of perovskite film is increased by 3 times when it is deposited on E-SnO₂ in comparison to that on the pristine SnO₂ (Fig. 2). The larger grain size can effectively suppress the moisture permeation

at grain boundaries,^{R21} resulting in improved environmental stability for the PSCs based on the E-SnO₂ ETLs. These explanations have been added in the manuscript (page 17).

Response references:

R1. Yang, D. et al. Surface optimization to eliminate hysteresis for record efficiency planar perovskite solar cells. *Energy Environ. Sci.* **9**, 3071-3078 (2016).

R2. Murgatroyd, P. N. Theory of space-charge-limited current enhanced by Frenkel effect. *J. Phys. D: Appl. Phys.* **3**, 151-156 (1970).

R3. Bube, R. H. Trap density determination by space-charge-limited currents. *J. Appl. Phys.* **33**, 1733 (1962).

R4. Zhao, W., Yao, Z., Yu, F., Yang, D. & Liu, S. Alkali metal doping for improved CH₃NH₃PbI₃ perovskite solar cells. *Adv. Sci.* **5**, 1700131 (2018).

R5. Dong, Q. et al. Electron-hole diffusion lengths > 175 μm in solution-grown CH₃NH₃PbI₃ single crystals. *Science* **347**, 967-970 (2015).

R6. Zhumekenov, A. A. et al. The role of surface tension in the crystallization of metal halide perovskites. *ACS Energy Lett.* **2**, 1782-1788 (2017).

R7. Zhao, H. et al. Enhanced stability and optoelectronic properties of MAPbI₃ films with cationic surface active agent for perovskite solar cells. *J. Mater. Chem. A*
DOI:10.1039/C8TA00457A.

R8. Li, P., Liang, C., Zhang, Y., Li, F. & Song, Y. Shao G Polyethyleneimine high-energy hydrophilic surface interfacial treatment toward efficient and stable perovskite solar cells. *ACS Appl Mater Interfaces* **30**, 32574-32580 (2016).

- R9. Wang, W. et al. Enhanced performance of $\text{CH}_3\text{NH}_3\text{PbI}_{3-x}\text{Cl}_x$ perovskite solar cells by $\text{CH}_3\text{NH}_3\text{I}$ modification of TiO_2 -perovskite layer interface. *Nanoscale Res. Lett.* **11**, 1-9 (2016).
- R10. Lee, H., Rhee, S., Kim, J., Lee, C. & Kim, H. Surface coverage enhancement of a mixed halide perovskite film by using an UV-ozone treatment. *J. Korean Phys. Soc.* **69**, 406-411 (2016).
- R11. Fu, P. et al. Efficiency improved for inverted polymer solar cells with electrostatically self-assembled BenMeIm-Cl ionic liquid layer as cathode interface layer. *Nano Energy* **13**, 175-282 (2015).
- R12. Majumder, S. Synthesis and characterisation of SnO_2 films obtained by a wet chemical process. *Mater. Sci.-Poland* **27**, 123-129 (2009)
- R13. Gundrizer, T. A. & Davydov, A. A. IR spectra of oxygen adsorbed on SnO_2 . *Reaction Kinetics and Catalysis Letters* **3**, 63-70 (1975).
- R14. Li, M. et al. High-efficiency robust perovskite solar cells on ultrathin flexible substrates. *Nature Commun.* **7**, 10214 (2016).
- R15. Makuta, S. et al. Photo-excitation intensity dependent electron and hole injections from lead iodide perovskite to nanocrystalline TiO_2 and spiro-OMeTAD. *Chem. Commun.* **52**, 673-676 (2016).
- R16. Zardetto, V., Brown, T. M., Reale, A. & Carlo, A. D. Substrates for flexible electronics: A practical investigation on the electrical, film flexibility, optical, temperature, and solvent resistance properties. *J. Polym. Sci. Pol. Phys.* **49**, 638-648 (2011).
- R17. An, Q. et al. High performance planar perovskite solar cells by ZnO electron transport layer engineering. *Nano Energy* **39**, 400-408 (2017).

R18. Chen, B. et al. Impact of capacitive effect and ion migration on the hysteretic behavior of perovskite solar cells. *J. Phys. Chem. Lett.* **6**, 4693-4700 (2015).

R19. Reenen, S. V., Kemerink, M. & Snaith, H. J. Modeling anomalous hysteresis in perovskite solar cells. *J. Phys. Chem. Lett.* **6**, 3808-3814 (2015).

R20. Heo, J. H. et al. Planar $\text{CH}_3\text{NH}_3\text{PbI}_3$ perovskite solar cells with constant 17.2% average power conversion efficiency irrespective of the scan rate. *Adv. Mater.* **27**, 3424-3430 (2015).

R21. Chu, Z. et al. Impact of grain boundaries on efficiency and stability of organic-inorganic trihalide perovskites. *Nat. Commun.* **8**, 2230 (2017).

Reviewers' Comments:

Reviewer #1:

Remarks to the Author:

The authors have done an exemplary job responding in detail to my, and the other reviewer's comments. The paper is now both more complete and more easily understandable. I think it will be an influential paper in the field. I therefore recommend publication of the manuscript in Nature Communications.

Reviewer #2:

Remarks to the Author:

Am satisfied with the answers. Two minor things:

1) where the electron goes down in energy, I would not use the term "barrier" but "offset"

2) The important thing is that the main text has been improved by incorporating the answers in the manuscript. The only answer which has not been incorporated in the text is the one below so I suggest incorporating in the main text to help any reader, not just me, understand why such a radius was used for the bending tests.

QUESTION: On page 17 the authors use a bending radius of 7mm. Why was this chosen? It is where ITO cracks? Please provide some references.

RESPONSE: According to a previous report,R16 it is safe for ITO to be bended to a radius of 14 mm, and when the bending radius is smaller than 14 mm, the ITO layer starts to crack, leading to significant degradation in conductivity. In order to examine the intrinsic mechanical stability of the flexible PSCs, we therefore adopted the bending radius of 7 mm, half of that suggested in literature, to test the flexible device.

Reviewer #3:

Remarks to the Author:

The quality of this manuscript has been greatly improved by the authors in this version. I would recommend the publication of this work in Nature Communications after the authors address the following critical issues.

1. As shown in Fig 1e, the overall transmittance of all the SnO₂-based ETLs is below 90 % across the wavelengths from 400-800 nm. Why does the EQE of the EDTA-SnO₂-based device (Fig. 4b) can exceed 90%?
2. Besides, the optical loss (or parasitic absorption) at the wavelengths from 400-550 nm induced by the SnO₂-based ETL is not reflected in the IPCE spectra. What is the possible reason? The authors need to elucidate this.

Responses to Reviews

Dear Editor and Reviewers:

Thank you very much for your insightful feedback on the manuscript.

We have carefully revised the manuscript to address all your comments and questions. In particular, we have conducted additional experiments to address some of your critical questions. All changes have been marked in blue color in the revised submission. A detailed point-by-point response is attached along with this letter.

We hope that all of you will find this revised submission satisfactory for publication in the “*Nature Communications*”. We very much look forward to hearing from you.

Sincerely,

Shashank Priya
Fellow, American Ceramic Society
President, Energy Harvesting Society

Point-By-Point Response to Referees' Comments

Reviewer #1 (Remarks to the Author):

The authors have done an exemplary job responding in detail to my, and the other reviewer's comments. The paper is now both more complete and more easily understandable. I think it will be an influential paper in the field. I therefore recommend publication of the manuscript in Nature Communications.

Response: We really thank the referee for recommending publication in Nature Communications.

Reviewer #2 (Remarks to the Author):

Am satisfied with the answers. Two minor things:

1) Where the electron goes down in energy, I would not use the term "barrier" but "offset"

Response: Thanks for your suggestion. We completely agree with this point that “offset” is more accurate than “barrier” to describe the electron goes down in energy, therefore we have revised the expression in the manuscript.

2) The important thing is that the main text has been improved by incorporating the answers in the manuscript. The only answer which has not been incorporated in the text is the one below so I suggest incorporating in the main text to help any reader, not just me, understand why such a radius was used for the bending tests.

QUESTION: On page 17 the authors use a bending radius of 7mm. Why was this chosen? It is where ITO cracks? Please provide some references.

Response: Thank you for your comments. According to a previous report (J. Polym. Sci. Pol. Phys., 2011, 49, 638-648), it is safe for ITO to be bended to a radius of 14 mm, and when the bending radius is smaller than 14 mm, the ITO layer starts to crack, leading to significant degradation in conductivity. In order to examine the mechanical stability of the flexible PSCs, we therefore adopted the bending radius of 14 mm, 12mm and 7 mm to test the flexible device. Fig. 7a shows device performance of the flexible solar cells measured after flexing for 500 times with different curvature radius, with the test procedure shown in Fig. 7a inset. It is clear that at the bending radius 14 mm, the device performance shows no observable degradation after the flexing test. When the bending radius is decreased to 12 mm and 7 mm, the PCE degraded to 17.82% and 16.84%, respectively. The reduced efficiency of the flexible PSCs is mainly caused by conductivity degradation of the ITO layer under the flexing stressing at small bending radius (J. Polym. Sci. Pol. Phys., 2011, 49, 638-648). We have added the contents into the manuscript (page 19 and reference No. 73)

Reviewer #3 (Remarks to the Author):

The quality of this manuscript has been greatly improved by the authors in this version. I would recommend the publication of this work in Nature Communications after the authors address the following critical issues.

Response: Thank you very much for referee's kind feedback and comments. We have conducted additional experiments and revised the manuscript to address all the questions.

1. As shown in Fig 1e, the overall transmittance of all the SnO₂-based ETLs is below 90 % across the wavelengths from 400-800 nm. Why does the EQE of the EDTA-SnO₂-based device (Fig. 4b) can exceed 90%?

Response: We characterized the reflection of ITO/E-SnO₂ and ITO/E-SnO₂/perovskite samples, as shown in Fig. R1. It can be seen that the average reflection value of ITO/E-SnO₂ is 12.21% in the wavelength range of 400-800 nm. However, when the perovskite absorber layer is deposited onto the ITO/E-SnO₂ substrate, the average reflection is significantly reduced to 6.88%. The reduced optical loss of the ITO/E-SnO₂/perovskite is often seen for multilayer coatings for the antireflection effect due to the difference refractive index between E-SnO₂ (~2.3) and perovskite (~2.9) (J. Sci. Techno., 2012, 4, 61-72; J. Phys.: Condens. Matter, 2017, 29, 245702). The smaller reflection leads to higher IPCE of the perovskite solar cells based on E-SnO₂ ETLs in the wavelengths from 400 nm to 800 nm. The results have been added in the manuscript and the Supplementary Information (page 14 and Fig. S11).

Figure R1. Reflection spectra of the glass/E-SnO₂ and glass/ITO/E-SnO₂/perovskite samples.

2. Besides, the optical loss (or parasitic absorption) at the wavelengths from 400-550 nm induced by the SnO₂-based ETL is not reflected in the IPCE spectra. What is the possible reason? The authors need to elucidate this.

Response: Thanks for your insightful question. The absorption spectrum of the FA_{0.95}Cs_{0.05}PbI₃ absorber was measured, as shown in Fig. R2. It is apparent that the absorption intensity is very strong in the wavelength range of 400-550 nm owing to large absorption coefficient of the perovskite in this wavelength range (Materials Today, 2015, 18, 65-72). It means that more photo-generated carriers would be produced in this range. Therefore, even though the glass/ITO/SnO₂ exhibits low transmittance in the wavelengths range of 400-550 nm, more photo-generated carriers make up the IPCE loss.

Figure R2. Absorption and transmittance spectra for FA_{0.95}Cs_{0.05}PbI₃ absorber and glass/ITO/SnO₂ sample, respectively.

Reviewers' Comments:

Reviewer #2:

Remarks to the Author:

All OK, please accept. However the numbering of the new reference in the text is 73 and in the manuscript is 72 so just make sure the numbering is changed to be correct

Reviewer #3:

Remarks to the Author:

The authors have addressed my concerns and I now recommend the acceptance of this work.

Point-By-Point Response to Referees' Comments

Reviewer #2 (Remarks to the Author):

All OK, please accept. However the numbering of the new reference in the text is 73 in the manuscript is 72 so just make sure the numebering is changed to be correct.

Response: Thanks for your insight comment. I am really sorry for the typo error. The numbering of this reference is 73. We have corrected it in the manuscript.

Reviewer #3 (Remarks to the Author):

The authors have addressed my concerns and I now recommend the acceptance of this work.

Response: Thank you very much for recommending publication.